# Spatial quality control bypasses cell-based limitations on proteostasis to promote prion curing

Courtney L Klaips[1,2¶], Megan L Hochstrasser[1†‡], Christine R Langlois[1†], Tricia R Serio[2]*

[1]Department of Molecular Biology, Cell Biology and Biochemistry, Brown University, Providence, United States; [2]Department of Molecular and Cellular Biology, University of Arizona, Tucson, United States

*For correspondence: tserio@email.arizona.edu

[†]These authors contributed equally to this work

Present address: [¶]Max Planck Institute of Biochemistry, Munich, Germany; [‡]Department of Molecular and Cell Biology, University of California, Berkeley, Berkeley, United States

Competing interests: The authors declare that no competing interests exist.

## Abstract

The proteostasis network has evolved to support protein folding under normal conditions and to expand this capacity in response to proteotoxic stresses. Nevertheless, many pathogenic states are associated with protein misfolding, revealing in vivo limitations on quality control mechanisms. One contributor to these limitations is the physical characteristics of misfolded proteins, as exemplified by amyloids, which are largely resistant to clearance. However, other limitations imposed by the cellular environment are poorly understood. To identify cell-based restrictions on proteostasis capacity, we determined the mechanism by which thermal stress cures the [$PSI^+$]/Sup35 prion. Remarkably, Sup35 amyloid is disassembled at elevated temperatures by the molecular chaperone Hsp104. This process requires Hsp104 engagement with heat-induced non-prion aggregates in late cell-cycle stage cells, which promotes its asymmetric retention and thereby effective activity. Thus, cell division imposes a potent limitation on proteostasis capacity that can be bypassed by the spatial engagement of a quality control factor.

## Introduction

The proper folding of proteins is essential to cellular homeostasis, and an extensive collection of protein quality control (PQC) pathways, known as the proteostasis network, has evolved to protect nascent and metastable proteins from misfolding and to reactivate or remove proteins that have already misfolded (*Powers et al., 2009*; *Wolff et al., 2014*). The PQC network is tailored to buffer protein folding in a distinct homeostatic niche but can adapt when these buffering thresholds are exceeded by elevating the expression of PQC factors, including proteases and molecular chaperones, to clear accumulating misfolded proteins (*Morimoto, 2008*; *Powers et al., 2009*). In cases such as thermal stress, these corrections are sufficient to restore balance, but in others such as aging, misfolded proteins assemble into ordered amyloid aggregates, which persist and dramatically alter cellular physiology by inducing disease (*Tuite and Serio, 2010*; *Voisine et al., 2010*; *Taylor and Dillin, 2011*; *Kim et al., 2013*). This proteostasis collapse has been linked to the unique ability of amyloids to incorporate and conformationally convert like protein to the misfolded state and to their high thermodynamic stability (*Chiti and Dobson, 2006*; *Jahn and Radford, 2008*). Together, these properties are thought to enhance the production and restrict the resolution of the misfolded protein to the point that the buffering capacity and adaptability of the proteostasis network is chronically exceeded.

Despite this natural upper boundary on proteostasis capacity, the heterologous overexpression of molecular chaperones in *Caenorhabditis elegans*, mice, *Drosophila*, yeast, and tissue culture-cell models of amyloidoses reduces proteotoxicity (*Chernoff et al., 1995*; *Morimoto, 2008*; *Broadley and Hartl, 2009*; *Holmes et al., 2014*). While these observations are often interpreted as evidence of

**eLife digest** Proteins must fold into specific shapes to work inside cells, and the misfolding of proteins is associated with a growing number of diseases. For example, prions are misfolded proteins that form insoluble aggregates called amyloids. These aggregates are not easily destroyed and can cause other nearby proteins to misfold and join the amyloid. This process of amyloid assembly leads to progressive diseases such as mad cow disease, Huntington's disease, Alzheimer's disease, and Parkinson's disease, which are collectively known as amyloidoses.

A series of biological pathways called the proteostasis network control protein integrity in a cell. Under normal conditions or even mildly stressful conditions—such as at slightly increased temperatures—the proteostasis network is able to prevent proteins from misfolding. However, if a cell is placed under lots of stress this network may become overwhelmed and misfolded proteins can accumulate. To date, the proteostasis network has not been linked to the clearance of amyloids.

A protein called Sup35, which is found in budding yeast, can exist as two different prion forms. Previous studies have shown that briefly heating the yeast cells can 'cure' the so-called 'weak' form of the prion. The 'strong' prion form, however, was thought to be unaffected by elevated temperature. These previous studies had only tested yeast cells that had been dividing for a few generations; it was unknown if cells that had been dividing for longer might respond differently.

Klaips et al. found that a protein called Hsp104—which helps to fold proteins properly—can break down the amyloid aggregates. This protein is normally only present in small amounts, but heating causes the levels of Hsp104 to rise. Klaips et al. found that the extra Hsp104 protein associated with the aggregates and led to their disassembly. When Hsp104 was prevented from associating with the prions, the aggregates were not cured even if high levels of Hsp104 were present in the cell.

When budding yeast form new cells, a daughter cell 'buds' off from the mother cell. Klaips et al. found that mother cells exposed to heat retain most of the Hsp104 when the cell divides, and this retention allowed Hsp104 to accumulate to a level required for the breakdown of amyloid aggregates. Therefore, under normal conditions, amyloids persist because cell division keeps the amount of Hsp104 below this threshold.

Previously it had been thought that the physical characteristics of amyloids accounted for their resilience in the face of the cell mechanisms designed to counteract protein misfolding. However, Klaips et al. show that the balance of the different mechanisms involved in proteostasis can be manipulated to create environments where amyloids are either created and maintained or destroyed. Targeting these mechanisms could therefore present new treatment options for amyloidosis.

amyloid resolution, existing protein has not been demonstrated to transition from an amyloid to a non-amyloid form in any of the studies. Instead, two correlations have been observed where the reduced proteotoxicity has been linked to a change in amyloid state. Either amyloid accumulation is enhanced by chaperone overexpression (*Douglas et al., 2009*; *Cushman-Nick et al., 2013*), or amyloid accumulation is reduced. In the few cases where the mechanism has been determined, the reduction in amyloid accumulation results from an inhibition of amyloid assembly by the overexpressed chaperone (*Kobayashi et al., 2000*; *Schaffar et al., 2004*; *Tam et al., 2006*; *Shorter and Lindquist, 2008*; *Winkler et al., 2012*). Thus, even the specific overexpression of individual chaperones is unable to extend the proteostasis upper boundary in vivo to resolve protein amyloids.

Although these targeted interventions have yet to succeed, studies conducted under conditions that reduce amyloid amplification indicate that amyloid clearance may not represent an insurmountable obstacle. For example, repressing expression of an amyloidogenic protein can reverse established toxicity and, at least in some cases, clear existing amyloid (*Yamamoto et al., 2000*; *Mallucci et al., 2003*; *Lim et al., 2011*). In addition, expression of a dominant-negative mutant also promotes disassembly of wild-type amyloid in vivo (*DiSalvo et al., 2011*). Together, these observations suggest that amyloid clearance mechanisms exist in vivo, and indeed amyloid resolution is biochemically feasible in vitro using purified chaperones such as yeast Hsp104, alone or in combination with its co-chaperones Hsp40, Hsp70, and small heat shock proteins (*Inoue et al., 2004*; *Shorter and Lindquist, 2004*,

*Lo Bianco et al., 2008*; *Shorter and Lindquist, 2008*). What limitations, then, restrict the ability of cells to expand proteostasis capacity to effectively resolve continuously expressed wild-type protein amyloids in vivo?

To identify cell-based limitations on proteostasis capacity, we focused on the mechanisms controlling persistence of the yeast prion [*PSI+*], the alternative, self-templating, amyloid form of the Sup35 protein (*Cox, 1965*; *Patino et al., 1996*; *Paushkin et al., 1996*; *Glover et al., 1997*; *King et al., 1997*). In this study, we report that a transient thermal stress surprisingly leads to the complete disassembly of existing Sup35 amyloid. This process requires the accumulation of heat-induced non-prion protein aggregates in cells primarily at the later stages of the cell cycle. The engagement of Hsp104 with these substrates, and its inability to resolve them before cell division, leads to asymmetric retention of the chaperone in cells that experienced the thermal stress. As a result, Hsp104 accumulates to a level that is sufficient to resolve amyloid aggregates. Thus, the kinetics of substrate engagement by a PQC factor and its partitioning during cell division impose cell-based limitations on proteostasis capacity.

## Results

### Sup35 amyloid is resolubilized by Hsp104 following thermal stress

Under normal growth conditions, [*PSI+*] propagates faithfully (*Cox, 1965*; *Derkatch et al., 1996*). However, at elevated temperatures where the PQC capacity is increased, [*PSI+*] becomes destabilized in a Sup35 conformation-specific manner. For example, the more thermodynamically stable but less efficiently propagated [*PSI+*]$^{Weak}$ variant is quantitatively 'cured' (i.e. converted to the non-prion [*psi–*] state) at elevated temperature in comparison with [*PSI+*]$^{Strong}$, whose propagation is unaltered under the same conditions (*Cox et al., 1988*; *Derkatch et al., 1996*; *Jung et al., 2000*; *Tanaka et al., 2006*; *Newnam et al., 2011*). This curing of [*PSI+*]$^{Weak}$ was linked to the inhibition of the molecular chaperone Hsp104 (*Newnam et al., 2011*), an observation that is seemingly counter to the idea that proteostasis capacity increases in response to stress (*Morimoto, 2011*). However, in this study, stationary phase cultures were only briefly diluted into fresh medium to re-establish exponential growth before exposure to elevated temperature (*Newnam et al., 2011*). Because stationary phase alters chaperone expression and blocks [*PSI+*] curing at elevated temperature (*Gasch et al., 2000*; *Newnam et al., 2011*), residual effects from the growth phase switch could alter the interaction between Sup35 aggregates and PQC factors. Therefore, we revisited the effects of elevated temperature on [*PSI+*] propagation, beginning with exponentially growing cultures.

To monitor transitions from the prion [*PSI+*] to the non-prion [*psi–*] state, we used yeast strains encoding a premature termination codon (PTC) in the *ADE1* gene. In [*PSI+*] strains, Sup35 is functionally compromised, leading to stop-codon read-through and the formation of white or pink colonies on rich medium, but in [*psi–*] strains, termination is faithful at the PTC, leading to the formation of red colonies on rich medium (*Chernoff et al., 1995*). Transiently elevating the growth temperature from 30°C to 40°C had no effect on viability (*Figure 1—figure supplement 1A*) or on [*PSI+*]$^{Strong}$ propagation (*Figure 1A*) but induced [*PSI+*]$^{Weak}$ curing (*Figure 1A,B*). Notably, both fully red and sectored colonies were observed, indicating that curing happened during both the thermal stress and subsequent recovery (*Figure 1B*). Thus, [*PSI+*]$^{Weak}$ propagation is similarly sensitive to elevated temperature in exponentially growing cultures and in those that have recently exited stationary phase.

At the normal growth temperature, large Sup35 aggregates are fragmented into smaller complexes by Hsp104 (*Chernoff et al., 1995*; *Eaglestone et al., 2000*; *Ness et al., 2002*; *Satpute-Krishnan et al., 2007*; *Kawai-Noma et al., 2009*). In a culture that recently exited stationary phase, the size of SDS-resistant Sup35 aggregates increased, as assessed by semi-denaturing detergent agarose gel electrophoresis (SDD-AGE) (*Kryndushkin et al., 2003*), following incubation at 40°C and a 2 hr recovery at 30°C (*Figure 1—figure supplement 1B*, left) (*Newnam et al., 2011*), consistent with an inhibition of fragmentation (*Newnam et al., 2011*). In contrast, SDS-resistant Sup35 aggregates were immediately reduced in size (*Figure 1C*, *Figure 1—figure supplement 1B*) and completely lost after recovery (*Figure 1—figure supplement 1B*, right) following identical treatment of an exponentially growing [*PSI+*]$^{Weak}$ strain, a progression suggesting the resolution of existing Sup35 aggregates. To test this possibility, we incubated a [*PSI+*]$^{Weak}$ culture at 40°C, returned the culture to 30°C in the presence of cycloheximide to repress new protein synthesis, and monitored the conversion of existing Sup35 from the amyloid [*PSI+*] state (i.e. SDS-resistant) to the non-amyloid [*psi–*] state (i.e. SDS-sensitive) (*Serio et al., 2000*; *Satpute-Krishnan and Serio, 2005*). In a control culture at 30°C, very little

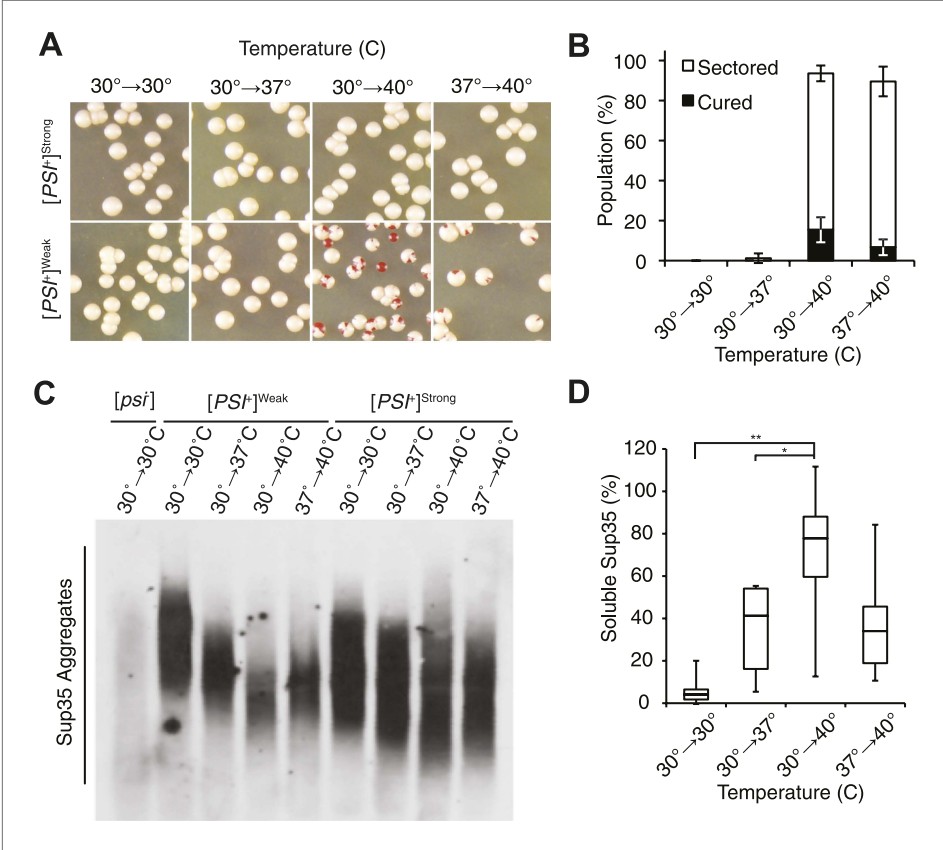

**Figure 1**. Thermal stress induces curing through resolution of Sup35 amyloid. (**A**) [*PSI*+]Strong (SLL2606) and [*PSI*+]Weak (SLL2600) cultures were incubated for 30 min at the indicated temperatures before plating on rich medium at 30°C to analyze curing by colony color phenotype, as described in the text. (**B**) Quantification of [*PSI*+]Weak (SLL2600) colony color phenotypes following treatment as described in (**A**). Colonies were scored as completely [*psi*−] (black), or sectored (partially [*psi*−], white). Data represent averages; error bars represent standard deviations; n = 3. (**C**) Semi-native lysates of [*psi*−] (SLL2119), [*PSI*+]Weak (SLL2600), and [*PSI*+]Strong (SLL2606) cultures were analyzed by semi-denaturing detergent agarose gel electrophoresis (SDD-AGE) and immunoblotting for Sup35 after treatment as described in (**A**). (**D**) Sup35 released from amyoid aggregates in a [*PSI*+]Weak strain (SLL2600) following treatment as described in (**A**) and recovery at 30°C in the presence of cycloheximide was determined by treating lysates with 2% SDS at 53°C, followed by SDS-PAGE and quantitative immunoblotting for Sup35. Lines represent medians; boxes represents upper and lower quartiles, and whiskers represent maximum and minimum; n = 5; *p = 0.02, **p = 0.01 by paired t-test.

The following figure supplement is available for figure 1:

**Figure supplement 1**. Characterization of thermal stress effects.

pre-existing Sup35 transitioned to an SDS-sensitive state despite the inhibition of new protein synthesis (*Figure 1D*), as expected (*DiSalvo et al., 2011*). However, following incubation at 40°C, over 70% of SDS-resistant Sup35 became detergent sensitive during recovery at 30°C (*Figure 1D*), indicating disassembly of existing Sup35 amyloid. To determine if the prion curing resulting from this disassembly was mediated by Hsp104, we chemically inhibited this factor with guanidine HCl (GdnHCl) treatment or reduced its dosage by creating a heterozygous disruption in a diploid strain (*Eaglestone et al., 1999*; *Jung and Masison, 2001*; *Grimminger et al., 2004*; *Kummer et al., 2013*; *Tariq et al., 2013*; *Zeymer et al., 2013*), and in both cases, [*PSI*+]Weak curing was reduced by more than 50% relative to the wild-type untreated strain (*Figure 2A,B*). Thus, Hsp104 promotes the disassembly of existing Sup35 amyloid in a [*PSI*+]Weak strain following thermal stress.

At elevated temperature, we noted that the size of SDS-resistant Sup35 aggregates is reduced in a [*PSI*+]Strong strain (*Figure 1C*), although curing does not occur (*Figure 1A*). Because [*PSI*+]Strong

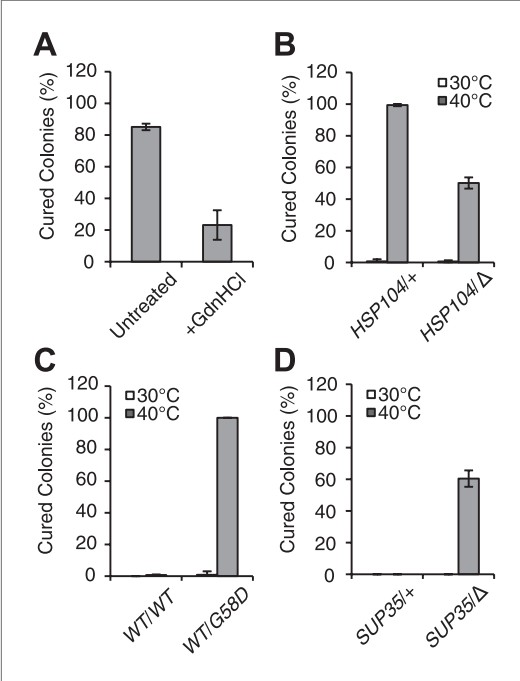

**Figure 2**. Curing is mediated by Hsp104 and depends upon propagation efficiency. (**A**) [*PSI*+]Weak cultures (SLL2600) were incubated at 40°C for 30 min in the absence (untreated) or presence of guanidine HCl (GdnHCl) and plated on YPD to quantify prion loss by colony color phenotype. Data represent means; error bars represent standard deviations; n = 3; p = 0.0004 by unpaired t-test. (**B**) A WT (*HSP104*/+; SY945) and a heterozygous disruption (*HSP104*/Δ; SY591) [*PSI*+]Weak diploid strain were incubated at 40°C for 90 min and plated on YPD to quantify prion loss by colony color phenotype. Data represent means; error bars represent standard deviations; n = 3; p < 0.0001 by unpaired t-test. (**C**) [*PSI*+]Strong strains expressing an extra copy of either WT (SY1646) or G58D (SY1648) Sup35 were incubated at 40°C for 90 min and plated on YPD to quantify prion loss by colony color phenotype. Data represent means; error bars represent standard deviations; n = 4; p < 0.0001 by unpaired t-test. (**D**) A WT (*SUP35*/+; SLL3071) and a heterozygous disruption (*SUP35*/Δ; SY957) diploid [*PSI*+]Strong strain were incubated at 40°C for 90 min and plated on YPD to quantify prion loss by colony color phenotype. Data represent means; error bars represent standard deviations; n = 3; p < 0.0001 by unpaired t-test.

propagates more efficiently than [*PSI*+]Weak, the former may be protected from curing at elevated temperature if the rate of Sup35 assembly continued to outpace the rate of its disassembly, a scenario that should be reversed by reducing the efficiency of [*PSI*+]Strong propagation. To test this idea, we subjected [*PSI*+]Strong diploid strains heterozygous for either a Sup35 mutant (G58D) or for a Sup35 disruption, which both reduce propagation efficiency (*Derdowski et al., 2010*; *DiSalvo et al., 2011*), to thermal stress. At 30°C, [*PSI*+] propagation is stable in both of these strains (*Figure 2C,D*); however at 40°C, both were now efficiently cured (e.g. ~100% for WT/*G58D*, ~60% for *SUP35*/Δ) (*Figure 2C,D*). These observations not only provide additional support for Sup35 amyloid disassembly as the mechanism of prion curing in response to thermal stress but also reveal that the inability of chaperones to resolve amyloid in vivo results from both the physical characteristics of these aggregates and cell-based limitations, which are bypassed in the distinct proteostasis niche created at elevated temperature.

## The asymmetric retention of Hsp104 is required for curing

Elevated temperature induces protein misfolding, and the cell responds to this stress by elevating the expression of PQC factors (*Morimoto, 2011*). To deconvolute the contributions of each of these events to [*PSI*+]Weak curing, we took advantage of the fact that we could modulate the efficiency of curing with variations in temperature. For example, while exposure to 40°C induced quantitative [*PSI*+]Weak curing, pretreatment at 37°C prior to exposure to 40°C slightly reduced curing (*Figure 1A,B*, compare proportion of fully cured colonies), and incubation at 37°C did not induce curing at all (*Figure 1A,B*). This failure to destabilize [*PSI*+]Weak at 37°C corresponded to an increase in aggregate size (*Figure 1C*) and a decrease in Sup35 solubilization (*Figure 1D*) relative to growth at 40°C alone, indicating a temperature-dependent modulation of amyloid resolution.

To determine the molecular basis of these differences in curing efficiency, we first monitored the levels of Sup35, Hsp104, Ssa1/2 (Hsp70), and Sis1 (Hsp40) proteins, which have all been implicated in Sup35 amyloid fragmentation (*Cox, 1965*; *Chernoff et al., 1995*; *Song et al., 2005*; *Higurashi et al., 2008*; *Tipton et al., 2008*; *Derdowski et al., 2010*). By quantitative immunoblotting, neither Sup35 (*Figure 3—figure supplement 1A*) nor chaperone levels (*Figure 3A*, *Figure 3—figure supplement 1B*) correlated with curing efficiency (*Figure 1A*), indicating that [*PSI*+]Weak curing could not be explained by simple changes in protein expression. Indeed, the specific overexpression of Hsp104 alone from a galactose-inducible promoter to levels that parallel those achieved during thermal stress (*Figure 3A* and *Figure 3—figure supplement 1C*) induces ~40% [*PSI*+]Weak curing (*Figure 3—figure supplement 1D*, 1.5 gen) in comparison with the ~95% [*PSI*+]Weak curing induced by thermal stress

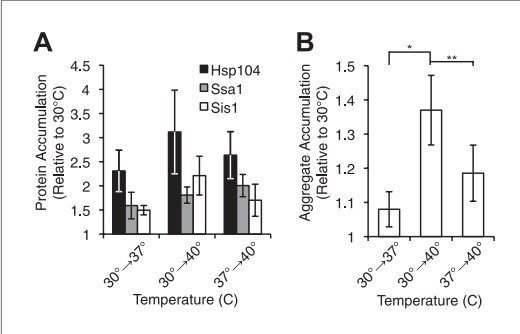

**Figure 3**. Heat-induced aggregate accumulation but not chaperone levels correlate with temperature. (**A**) A [*PSI*⁺]^Weak strain (SLL2600) was incubated at 30°C, 37°C, 40°C, or 37°C before 40°C for 30 min, and lysates were prepared and analyzed by SDS-PAGE and quantitative immunoblotting for Hsp104 (black), Ssa1 (gray), and Sis1 (white). Data represent means; error bars represent standard deviations; n ≥ 3. (**B**) Aggregates from lysates of a [*PSI*⁺]^Weak strain (SLL2600) following treatment as described in (**A**) were prepared and analyzed by differential centrifugation and Bradford assay. Data represent means; error bars represent standard error; n = 6; *p = 0.0014, **p = 0.0052 by paired t-test.

The following figure supplement is available for figure 3:

**Figure supplement 1**. Effects of thermal stress and Hsp104 on protein accumulation.

(*Figure 1B*) (*DiSalvo et al., 2011*, *Wegrzyn et al., 2001*). Moreover, Hsp104 overexpression alone leads to an increase in the size of SDS-resistant Sup35 aggregates isolated from a [*PSI*⁺]^Weak strain (*Kryndushkin et al., 2003*), as previously reported for [*PSI*⁺]^Strong (*Figure 3—figure supplement 1E*) (*Kryndushkin et al., 2003*), but this outcome is in obvious contrast to the disassembly of Sup35 amyloid that we observe upon thermal stress (*Figure 1C,D*). Thus, thermal stress and chaperone overexpression induce distinct changes in prion propagation.

We next assessed the accumulation of misfolded proteins following shifts in temperature to determine if this event correlated with [*PSI*⁺]^Weak curing efficiency. By differential centrifugation, protein aggregates accumulated independent of prion status at all elevated temperatures (*Figure 3B*, *Figure 3—figure supplement 1F*), but in contrast to chaperone expression (*Figure 3A*, *Figure 3—figure supplement 1B*), the severity of this accumulation was impacted by growth temperature. At 37°C, protein aggregation increased by less than 10% in comparison with a culture maintained at 30°C (*Figure 3B*, column 1), but in cultures treated at 37°C followed by 40°C or directly at 40°C, this level rose to ~20% or ~40%, respectively (*Figure 3B*, columns 3 and 2). Thus, the accumulation of protein aggregates (*Figure 3B*) correlates directly with curing efficiency at the various temperatures (*Figure 1A*).

We noted, however, that this correlation was not observed for a [*PSI*⁺]^Weak culture treated with GdnHCl during a 40°C incubation, which strongly reduced curing efficiency (*Figure 2A*) but did not reduce the accumulation of protein aggregates (*Figure 3—figure supplement 1G*). Nevertheless, numerous studies have reported the localization of chaperones to cytoplasmic quality control foci upon exposure to proteotoxic stresses (*Aguilaniu et al., 2003*; *Erjavec et al., 2007*; *Kaganovich et al., 2008*; *Specht et al., 2011*; *Wolfe et al., 2013*), and GdnHCl blocks the association of Hsp104 with at least one substrate (*Winkler et al., 2012*). To determine if Hsp104 localization to heat-induced aggregates rather than their accumulation per se determined prion-curing efficiency, we replaced endogenous *HSP104* with an *HSP104-GFP* fusion, which supports [*PSI*⁺] propagation (*Figure 4—figure supplement 1A*). At 40°C, this strain exhibited time-dependent [*PSI*⁺]^Weak curing (*Figure 4—figure supplement 1B*) and accumulated protein aggregates (*Figure 4—figure supplement 1C*) and Hsp104-GFP to wild-type levels, albeit with slightly delayed kinetics (*Figure 4—figure supplement 1D*). At elevated temperatures, we observed an increase Hsp104-interacting proteins as assessed by co-immunocapture (*Figure 4A*) and the localization of Hsp104-GFP to cytoplasmic foci (*Figure 4B,C*), which also contain the model substrate firefly luciferase-mCherry (*Figure 4—figure supplement 1E*). The amount of co-immunocaptured proteins (*Figure 4A* [2.5-fold increase at 37°C and 4.2-fold increase at 40°C relative to 30°C]) and the number and intensity of Hsp104-GFP fluorescent foci (*Figure 4C*) corresponded to both the accumulation of heat-induced protein aggregates (*Figure 3B*) and the efficiency of curing (*Figure 1A*). Notably, the Hsp104-GFP fluorescence pattern was unaltered in a non-prion [*psi*⁻] strain (*Figure 4B*), indicating that Hsp104-GFP was engaged with non-prion substrates. Treatment of a [*PSI*⁺]^Weak culture with GdnHCl during an incubation at 40°C, which strongly reduces Hsp104-GFP association with heat-induced interacting proteins (*Figure 4A* [1.7-fold decrease relative to 40°C in the absence of GdnHCl]) and localization to cytoplasmic foci (*Figure 4—figure supplement 1F*), also reduces the efficiency of curing (*Figure 2A*, *Figure 4—figure supplement 1G*). Thus,

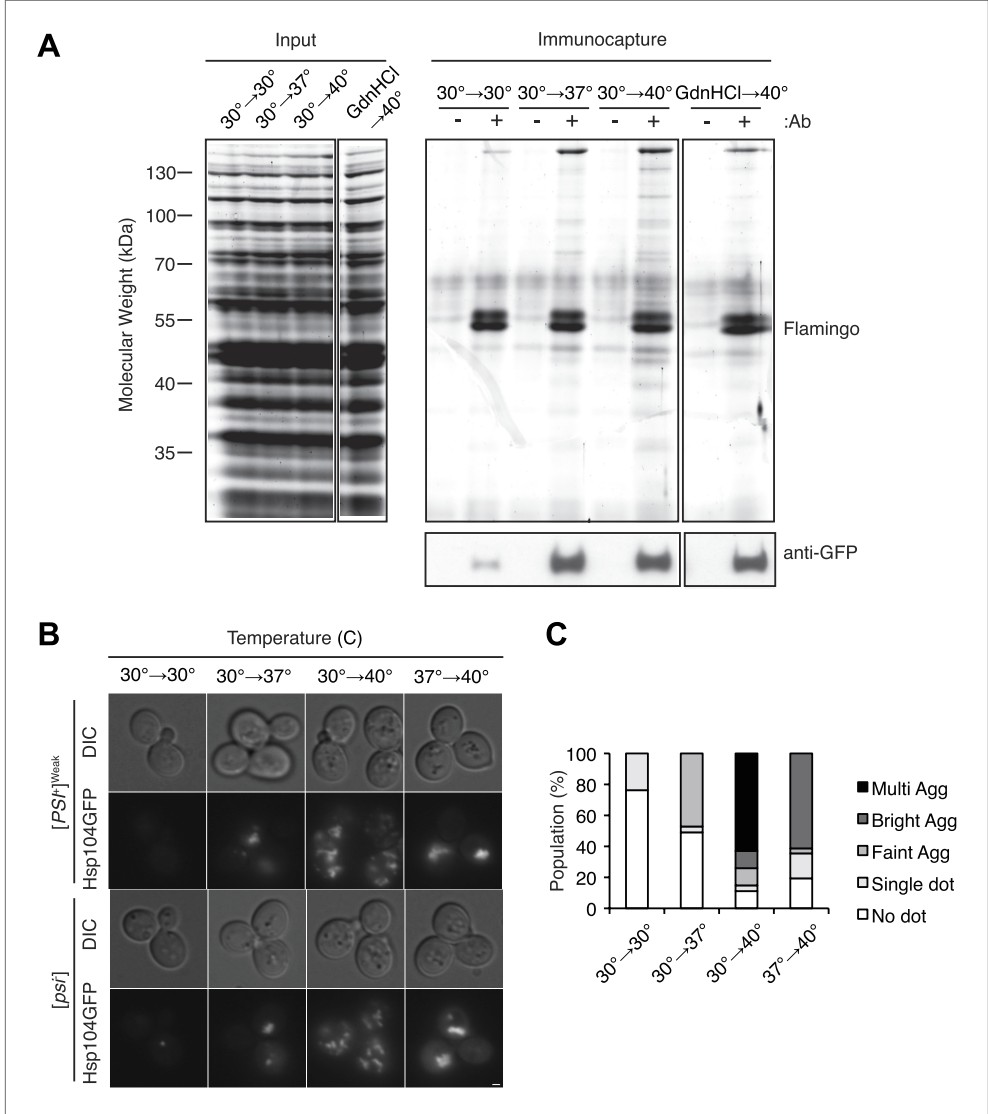

**Figure 4**. Hsp104 engages heat-induced substrates upon thermal stress. (**A**) A [*PSI*+]Weak strain with a GFP-tagged endogenous Hsp104 (SY2126) was incubated at 30°C, 37°C, 40°C, or 40°C with GdnHCl for 30 min, and immuno-capture in the presence (+) or absence (−) of anti-GFP antibodies (Ab) was performed on native lysates. Proteins were analyzed by SDS-PAGE and general protein staining (Flamingo, top), or immunoblotting for GFP (bottom). (**B**) [*PSI*+]Weak (SY2126) or [*psi*−] (SY2125) *HSP104GFP* strains were incubated at 30°C, 37°C, 40°C, or 37°C before 40°C for 90 min, and the pattern of Hsp104-GFP fluorescence was examined by microscopy. Scale bar = 1 μm. (**C**) Quantification of Hsp104-GFP fluorescence pattern in [*PSI*+]Weak (SY2126) cells, treated as described in (**B**): no localization (white); single dot (light gray); faint aggregate (medium gray); bright aggregate (dark gray); multiple bright aggregates (black); n > 25.

The following figure supplement is available for figure 4:

**Figure supplement 1**. Characterization of HSP104GFP strain.

the specific engagement of Hsp104 with heat-induced aggregates, rather than simply their presence, correlates with curing at elevated temperature.

How does this chaperone engagement with heat-induced aggregates lead to the resolution of Sup35 amyloid? One possibility is that the asymmetric localization of Hsp104, resulting from its engagement with heat-induced protein aggregates (*Erjavec et al., 2007*), increases its accumulation in a subpopulation of cells beyond that which can be achieved by its transcriptional up-regulation. To

test this possibility, we first monitored the partitioning of Hsp104-GFP during cell division following incubation at various temperatures using microfluidics and fluorescence microscopy. Starting with budded cells, mother cells accumulated ~60% of Hsp104-GFP following the completion of cell division at 30°C (*Figure 5A*, gray), which is comparable to the accumulation of untagged GFP expressed from the same promoter (*Figure 5—figure supplement 1A*) and thus likely reflects the volume differences between mother and daughter cells. This baseline asymmetry progressively increased as the temperature was increased to 37°C (~65% retention), 37°C followed by 40°C (~73% retention), and finally 40°C (~75% retention; *Figure 5A*, gray). Notably, both Ssa1-GFP and Sis1-GFP fusions also localized to cytoplasmic, and, in the case of Sis1, nuclear foci (*Figure 5—figure supplement 1B,C*), but neither was asymmetrically retained following incubation at 40°C (*Figure 5—figure supplement 1D,E*), although their levels were elevated relative to 30°C (*Table 1*) due to their enhanced expression (*Figure 3A*). Thus, curing efficiency (*Figure 1A,B*) correlates directly with the asymmetric retention of Hsp104 in cells at elevated temperature.

To determine if this correlation was a requirement, we next disrupted the asymmetric retention of Hsp104 and determined its effects on curing. Disruption of the formin *BNI1* (*Kohno et al., 1996*) did not alter Hsp104 expression levels or the accumulation of protein aggregates at 30°C and 40°C relative to a wild-type strain, (*Figure 5—figure supplement 1F,G*) but, Hsp104 asymmetric retention was reduced (*Figure 5B*, green), as expected (*Liu et al., 2010*). Strikingly, curing was dramatically suppressed from ~80% for a wild-type strain to ~10% in the Δ*bni1* strain (*Figure 5C*). Likewise, GdnHCl treatment before thermal stress, which blocked both Hsp104 engagement with heat-induced aggregates (*Figure 4—figure supplement 1F*) and curing at elevated temperature (*Figure 2A*, *Figure 4—figure supplement 1G*), also reduced Hsp104-GFP asymmetric retention following exposure to 40°C (*Figure 5D*). Thus, the asymmetric retention of Hsp104 is required for curing.

Our single-cell analyses of Hsp104-GFP partitioning indicated that a relatively minor change in chaperone retention from 65% to 75%, which corresponded to a 2.2-fold increase in accumulation based on fluorescence intensity (compare 37°C–40°C, *Table 1*, *Figure 5A*), correlated with a quantitative switch from prion stability to curing (*Figure 1A,B*), suggesting the existence of a biological threshold in this range. To determine directly if cells accumulating Hsp104-GFP corresponded to those cured of [*PSI+*]Weak, we incubated a [*PSI+*]Weak culture at 40°C and then isolated single unbudded cells on rich solid medium at 30°C. Following budding and cell division, mother and daughter cells were separated by micromanipulation and grown into colonies, which were then dispersed on rich solid medium to quantify prion retention. Mother cells, which experienced the elevated temperature and accumulated Hsp104 (*Figure 5A*), were more likely to be cured than their daughters (*Figure 5E*, note most data points fall below the diagonal), as predicted by our hypothesis. To more quantitatively correlate Hsp104-GFP accumulation with curing efficiency, we analyzed the distribution of Hsp104-GFP in a population of cells by flow cytometry. At 30°C, Hsp104-GFP fluorescence was distributed normally in the population (*Figure 5F*, dotted). Following incubation at 40°C, Hsp104-GFP fluorescence intensity in the population increased and its distribution was heterogeneous (*Figure 5F*, solid). When these subpopulations were separated by FACS and analyzed for colony-based phenotype, the efficiency of curing correlated directly with the accumulation of Hsp104-GFP (*Figure 5F,G*). Together, these observations indicate that cells exposed to elevated temperature accumulate heat-induced protein aggregates, asymmetrically retain Hsp104 in a manner that is proportional to these substrates, and ultimately cure [*PSI+*]Weak.

But, is Hsp104 enzymatic activity required for this curing, or is its asymmetric localization alone sufficient? As noted above, when cells are treated with GdnHCl before thermal stress, Hsp104 localization to cytoplasmic foci and asymmetric retention are both reduced (*Figure 5D*, *Figure 4—figure supplement 1F*). However, we reasoned the Hsp104 association with its substrates would be dynamic and modulated by its ATPase cycle. Indeed, blocking the ATPase activity of Hsp104 after thermal stress with a 90-min treatment with GdnHCl failed to reduce Hsp104-GFP localization to cytoplasmic foci (*Figure 4—figure supplement 1F*) or its asymmetric retention (*Figure 5D*), presumably because the chaperone bound to heat-induced substrates but was unable to release them once inhibited with GdnHCl. Despite the asymmetric localization of Hsp104-GFP under these conditions, [*PSI+*]Weak curing was reduced by nearly 50% (*Figure 4—figure supplement 1G*). Thus, both Hsp104 asymmetric localization and activity are required to induce [*PSI+*]Weak curing following thermal stress.

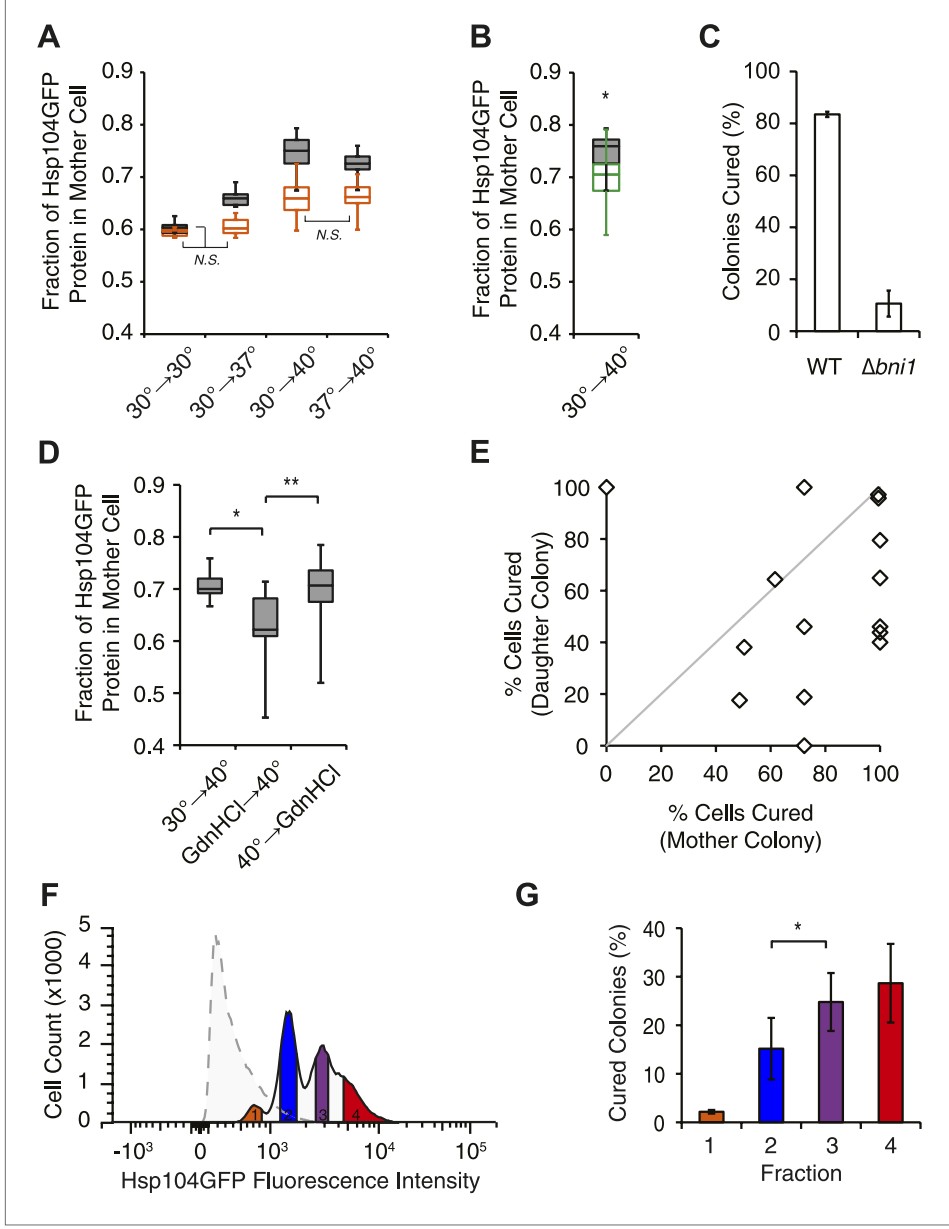

**Figure 5**. Curing results from the asymmetric localization of Hsp104 following thermal stress. (**A**) A [*PSI*+]^Weak *HSP104GFP* culture (SY2126) was imaged over time in a microfluidics chamber at 30°C after a 30 min incubation at 30°C, 37°C, 40°C, or 37°C before 40°C. Fluorescence intensity in daughter and mother cells was quantified at the first cell division in cells that were budded (gray) or unbudded (orange) after thermal stress. Lines represent medians; boxes represent upper and lower quartiles, and whiskers represent maximum and minimum. All pairwise comparisons are significantly distinct, with a p < 0.015, except where indicated (N.S.), by unpaired t-test; n ≥ 10. (**B**) A [*PSI*+]^Weak *HSP104GFP* WT (SY2126, gray) or *BNI1* deletion strain (Δ*bni1*) (SY2486, green) was imaged over time in a microfluidics chamber at 30°C after a 30 min incubation at 40°C. Fluorescence intensity in daughter and mother cells was quantified at the first cell division. Lines represent medians; boxes represent upper and lower quartiles; and whiskers represent maximum and minimum; n ≥ 14; p = 0.0075 by unpaired t-test. (**C**) [*PSI*+]^Weak WT (SLL2600) or Δ*bni1* strains (SY1888), treated as described in (**B**), were plated on YPD to analyze curing by colony color phenotype. Data represent means; error bars represent standard deviations; n = 3; p < 0.0001 by unpaired t-test. (**D**) A [*PSI*+]^Weak *HSP104GFP* strain (SY2126) was imaged over time in a microfluidics chamber at 30°C after a 30 min incubation at 40°C and with GdnHCl added before or after the 40°C incubation. Fluorescence intensity in daughter and mother cells was quantified at the first cell division. Lines represent medians; boxes represent upper and lower quartiles; and whiskers represent maximum and minimum; n > 11; *p = 0.0003, **p = 0.0026 by unpaired t-test. *Figure 5. Continued on next page*

*Figure 5. Continued*

(**E**) A [*PSI*⁺]^Weak strain (SLL2600) was incubated at 40°C for 30 min and plated on rich medium. Mother and daughter pairs were separated by micromanipulation and allowed to form colonies, which were then dispersed to YPD for analysis of curing by colony color phenotype. n = 15. (**F**) A [*PSI*⁺]^Weak *HSP104GFP* culture (SY2126) was incubated at 30°C (dotted) or at 40°C for 30 min and allowed to recover for 30 min at 30°C (solid) before analysis of GFP fluorescence intensity by flow cytometry. Based on these intensities, cells were sorted into four fractions (orange, blue, purple, red) by FACS. (**G**) Cells collected in (**F**) were plated on YPD to analyze curing by colony color phenotype. Data represent means; error bars represent standard deviations; n = 2; *p = 0.02 by paired t-test.

The following figure supplement is available for figure 5:

**Figure supplement 1**. Characterization of chaperone asymmetric retention following thermal stress.

## Cell-cycle stage and substrate-chaperone dynamics impact amyloid resolution

The distribution of Hsp104-GFP in a population of [*PSI*⁺]^Weak cells that had been exposed to 40°C was very complex in contrast to the normal distribution of Hsp104-GFP at 30°C (*Figure 5F*), suggesting that subpopulations of cells were differentially retaining the chaperone. One source of heterogeneity in the population was cell-cycle stage, as our experiments used asynchronous cultures (*Figure 6A*). To determine if cell-cycle stage at the time of thermal stress impacted Hsp104 partitioning and explained this heterogeneity, we arrested cells in G1 with α-factor or at the G2/M transition with nocodazole (*Amon, 2002*), exposed these cultures to 40°C incubation, and analyzed them by flow cytometry. Treatment with α-factor (*Figure 6B*) and nocodazole (*Figure 6C*) efficiently synchronized cultures at the non-budded or large-budded stages, respectively, and did not alter Hsp104 protein levels or localization relative to the asynchronous culture at 30°C (*Figure 6—figure supplement 1A,B*). At 40°C, Hsp104-GFP protein levels increased to similar extents in the asynchronous and arrested cultures (*Figure 6—figure supplement 1A*), and its localization to cytoplasmic foci was similar in all cases (*Figure 6—figure supplement 1B*). By flow cytometry, the distribution of Hsp104-GFP in the α-factor arrested culture remained normal (*Figure 6D*), but in the nocodazole-arrested culture, this distribution became bimodal (*Figure 6E*), indicating that Hsp104-GFP asymmetry is established immediately, even before cell division.

We next assessed the impact of cell-cycle stage on [*PSI*⁺]^Weak curing at elevated temperature. Arrest, without exposure to elevated temperature, did not induce curing (*Figure 6F,G*). In α-factor arrested cells, exposure to 40°C at release inefficiently cured [*PSI*⁺]^Weak (~15%; *Figure 6F*), but in nocodazole-arrested cells, curing was nearly quantitative (~90%; *Figure 6G*) consistent with the asymmetric localization of Hsp104-GFP in the latter but not the former case (*Figure 6D,E*). These observations suggest that cells at the end of the cell cycle are more sensitive to curing at elevated temperature than those at the beginning of the cell cycle. To test this idea, we released cultures from arrest and, after 30 min of growth at 30°C, exposed them to 40°C. For the culture originally arrested with α-factor, sensitivity to curing at elevated temperature increased (*Figure 6F*) as cells progressed into the late stages of the

**Table 1.** Relative fluorescence intensity in mother cells

| Treatment (°C) | Hsp104 (Relative to 30°C) | Ssa1 (Relative to 30°C) | Sis1 (Relative to 30°C) |
|---|---|---|---|
| 30°→30° | 1 ± 0.1 (24) | 1 ± 0.2 (29) | 1 ± 0.1 (18) |
| 30°→37° | 1.6 ± 0.2 (11) | | |
| 30°→40° | 3.5 ± 0.6 (52) | 2.7 ± 0.5 (18) | 1.5 ± 0.1 (7) |
| 37°→40° | 3.4 ± 0.4 (46) | | |

[*PSI*⁺]^Weak *HSP104GFP*(SY2126), *SSA1GFP* (SY2658), or *SIS1GFP* (SY2447) cultures were treated at indicated temperatures and were imaged over time at 30°C using microfluidics and fluorescence microscopy. Average fluorescence intensity in mother cells with indicated standard deviations (±), which originated from budded cells at the time of thermal stress, was measured at the first cell division. Number of cells analyzed is indicated in parentheses. p values are <0.001 for all comparisons to 30°C treatment.

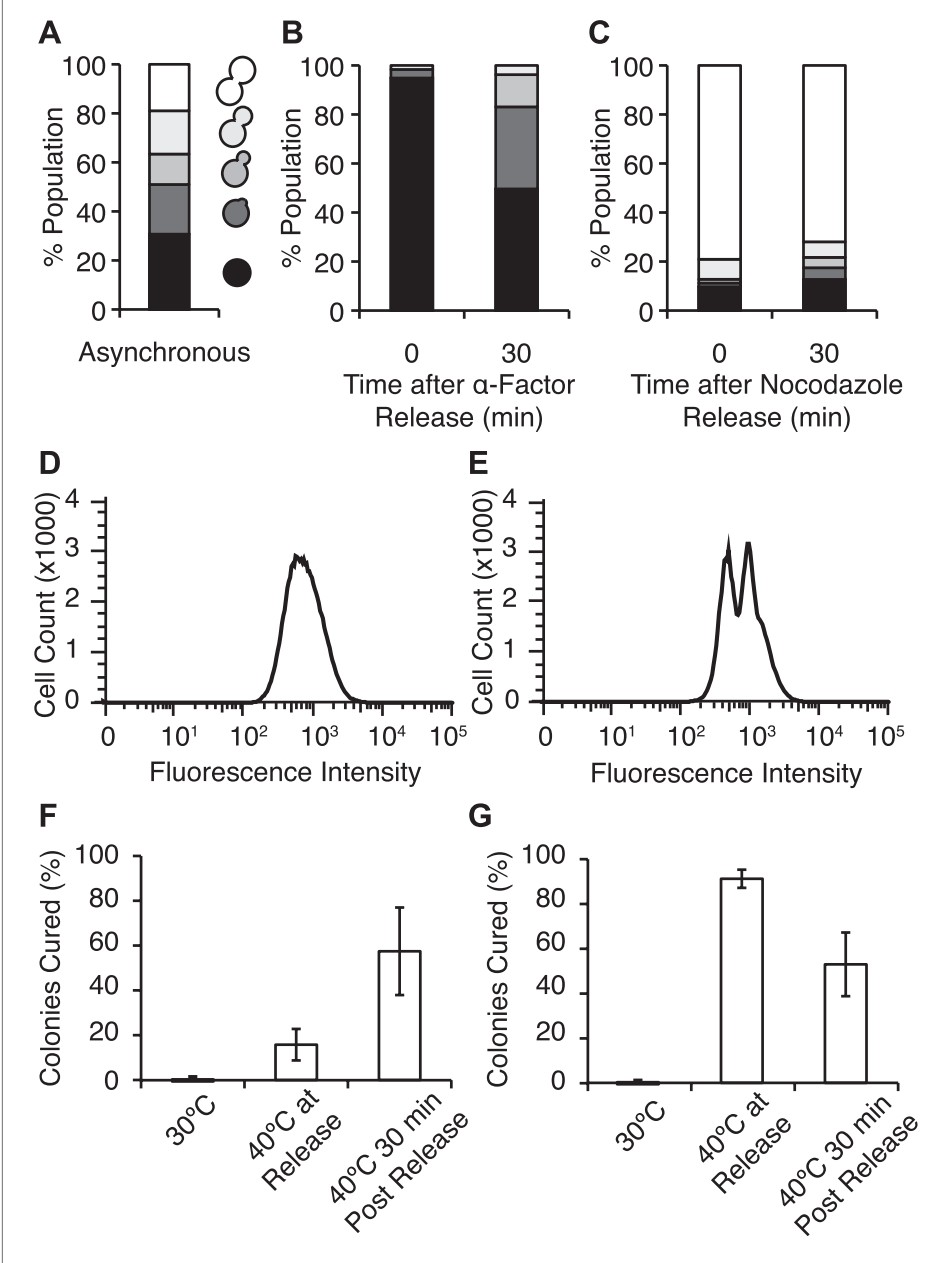

**Figure 6**. Efficient curing occurs in late cell-cycle staged cells following thermal stress. (**A**) Single cells from an asynchronous WT [*PSI*+]^Weak culture (SLL2600) were scored for morphology following bright-field imaging by microscopy: unbudded (black), tiny bud (dark gray), small bud (gray), medium bud (light gray), large bud (white). n = 153. (**B**) α-factor-arrested cultures were analyzed as in (**A**) over time after release. n ≥ 250. (**C**) Nocodazole-arrested cultures were analyzed as in (**A**) over time after release. n ≥ 175. (**D**) A [*PSI*+]^Weak *HSP104GFP* strain (SY2126) released from α-factor arrest was incubated at 40°C (solid black lines) for 30 min before analysis by flow cytometry. 100,000 cells were analyzed per sample. (**E**) A [*PSI*+]^Weak *HSP104GFP* strain (SY2126) released from nocodazole arrest was incubated at at 40°C (black lines) for 30 min before analysis by flow cytometry. 100,000 cells were analyzed per sample. (**F**) α-factor-arrested cultures (SLL2600) were incubated at 40°C for 30 min immediately or 30 min after release, and curing was quantified by colony color phenotype after plating on YPD at 30°C. Data represent means; error bars represent standard deviations; n = 3; p = 0.0255 by unpaired t-test. (**G**) Nocodazole-arrested cultures (SLL2600) were incubated at 40°C for 30 min immediately or 30 min after release, and curing was quantified colony color phenotype after

*Figure 6. Continued on next page*

*Figure 6. Continued*

plating on YPD at 30°C. Data represent means; error bars represent standard deviations; n = 3; p = 0.0263 by unpaired t-test.

The following figure supplement is available for figure 6:

**Figure supplement 1**. Characterization of chaperone accumulation and engagement in arrested cultures.

cell cycle (*Figure 6B*), and for the culture originally arrested with nocodazole, this sensitivity declined (*Figure 6G*) with cell-cycle progression (*Figure 6C*). Thus, curing occurs most efficiently when cells at a late stage of the cell cycle are exposed to elevated temperature.

Our earlier experiments linked curing to the asymmetric retention of Hsp104 at elevated temperature (*Figure 5*). To determine if cell-cycle stage impacts this asymmetry, we analyzed Hsp104-GFP distribution in mother–daughter pairs resulting from the growth and division of unbudded cells isolated from asynchronous cultures that were exposed to elevated temperatures. In comparison with budded cells, Hsp104-GFP retention was significantly reduced at all temperatures when unbudded cells were exposed to elevated temperature, but the magnitude of the effect was most severe for conditions that induced curing (30°C→40°C and 37°C→40°C; *Figure 5A*, orange), indicating a cell-cycle stage dependence on Hsp104-GFP retention at elevated temperature.

Because cell-cycle stage did not obviously alter the engagement of Hsp104-GFP with protein aggregates accumulating at elevated temperature (*Figure 6—figure supplement 1B*), the more efficient partitioning of Hsp104-GFP and the reduced curing in unbudded cells could reflect the resolution of heat-induced protein aggregates and thereby the release of Hsp104-GFP during the extended time before cell division in comparison with budded cells. Indeed, nearly 100% of cells contained Hsp104-GFP foci immediately after thermal stress (*Figure 7A*) but only ~80% still contained foci when cell division re-initiated ~150 min after incubation at 40°C (*Figure 7A*). Thus, the relative timing of substrate release and cell division could contribute to Hsp104 asymmetric retention and thereby curing. Consistent with this idea, 60% of unbudded cells, which are inefficiently cured (*Figure 6F*), resolved Hsp104-GFP foci prior to cell division (*Figure 7B* [165–210 min], *Figure 7C*), allowing the partitioning of the chaperone (*Figures 7B and 5A*, orange). In budded cells, which are efficiently cured (*Figure 6G*), only ~8% of cells had resolved heat-induced Hsp104-GFP foci by the time the cell divided (*Figure 7B* [105 min], *Figure 7C*), leading to the asymmetric retention of Hsp104-GFP (*Figures 7B and 5A*, gray). Together, these observations indicate that Hsp104 is retained in cells exposed to elevated temperature if it is unable to resolve its heat-induced substrates prior to cell division. Because sensitivity to curing at elevated temperature correlated with cell-cycle stage (*Figure 6F,G*) and Hsp104-GFP localization to these cytoplasmic foci (*Figure 5*), substrate–chaperone dynamics must create a temporal limitation on proteostasis capacity.

## Discussion

In *Saccharomyces cerevisiae*, expression of the molecular chaperone Hsp104, even at its low basal level, reduces organismal fitness at the normal growth temperature; however, survival at elevated temperatures is absolutely dependent on Hsp104, whose expression is induced to high levels by heat shock (*Sanchez et al., 1992*; *Escusa-Toret et al., 2013*). Thus, cell-based limitations must finely tune proteostasis capacity not only to control protein misfolding induced by stress but also to allow normal protein folding in the absence of these challenges (*Morimoto, 2008*). Using the yeast prion [*PSI*⁺] as a model to understand the in vivo interactions between amyloid and PQC pathways, we have uncovered one such pathway. While [*PSI*⁺]^Weak is mitotically stable at the normal growth temperature (~3% loss) (*Derkatch et al., 1996*), a transient sub-lethal thermal stress induces quantitative curing (*Figure 1* and *Figure 8*) through the disassembly of existing Sup35 amyloid by Hsp104 (*Figures 1, 2 and 8*). Our studies indicate that the increase in Hsp104 expression at elevated temperature alone is not sufficient to induce Sup35 amyloid resolution and [*PSI*⁺]^Weak curing (*Figures 1, 3*). Rather, Hsp104 must engage heat-induced protein aggregates for a period that exceeds the time to the next cell division (*Figures 7, 8*). As a result, Hsp104 is asymmetrically localized to the cells that experienced the thermal stress (*Figures 5, 8*), and this increase in chaperone accumulation, along with its activity, promotes curing in the same cells (*Figures 5, 8*). Thus, chaperone spatial engagement, substrate

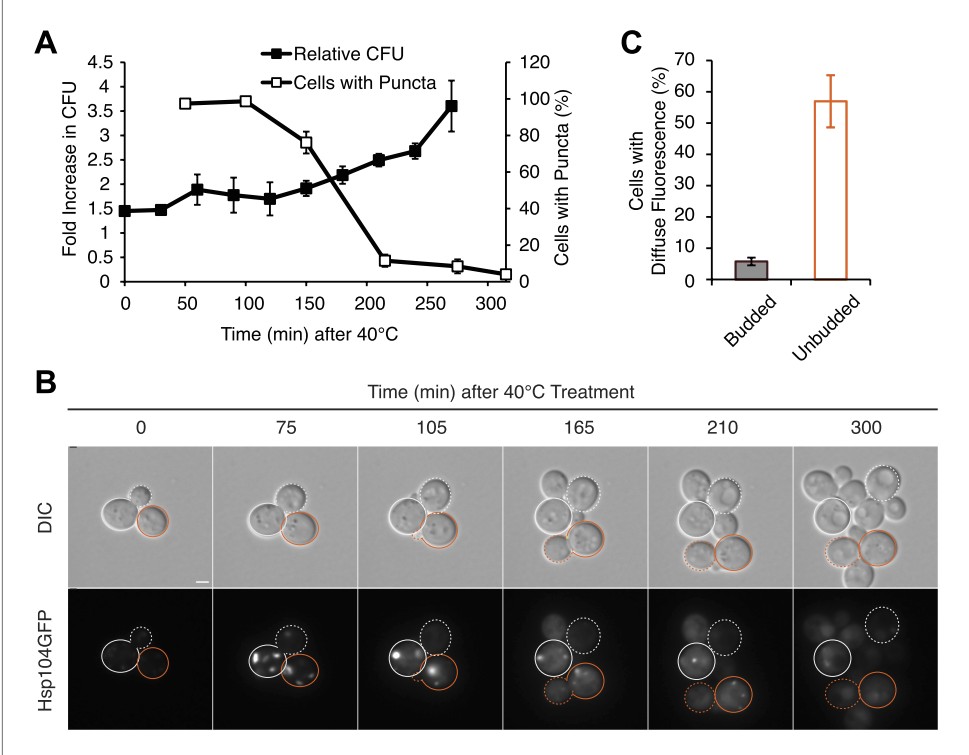

**Figure 7**. Substrate–chaperone engagement must exceed time to cell division to induce curing. (**A**) The number of [*PSI*⁺]^Weak *HSP104GFP* (SY2126) cells containing fluorescent foci was quantified in cultures recovering at 30°C over time following a 90 min incubation at 40°C (white). Colony forming units in these cultures were quantified by plating (black). Data represent means; error bars represent standard deviations; n = 3. (**B**) [*PSI*⁺]^Weak *HSP104GFP* cells (SY2126) treated for 30 min at 40°C and imaged over time in a microfluidics chamber are shown. Cells that were budded at the time of thermal stress are outlined in white, while unbudded cells are outlined in orange. Solid lines mark mothers, and dotted lines mark daughters. Scale bar = 1 μm. (**C**) A [*PSI*⁺]^Weak *HSP104GFP* strain (SY2126) was imaged over time in a microfluidics at 30°C after a 30 min incubation at 40°C chamber. Budded or unbudded cells were scored at the first cell division for the presence or absence of fluorescent aggregates. Data represent means; error bars represent standard deviations; n = 3; p = 0.0005 by unpaired t-test.

processing dynamics, and partitioning during cell division represent cell based limitations on proteostasis capacity.

Metazoans lack an Hsp104 homolog (*Torrente and Shorter, 2013*), but disaggregase activity has also recently been linked to a multi-component system in yeast comprised of Hsp110, Hsp70, and Hsp40, and this activity is conserved in the *C. elegans* and human homologs of these chaperones (*Shorter, 2011*; *Rampelt et al., 2012*; *Mattoo et al., 2013*). This system is largely ineffective in the disaggregation of amyloid in vitro (*Shorter, 2011*) but can promote the slow disassembly of amyloid from fiber ends in the presence of small heat shock proteins, such as Hsp26 and Hsp42 from yeast or HspB5 from humans (*Duennwald et al., 2012*). Like Hsp104 in yeast, Hsp110 localizes to foci containing misfolded protein in human cells following thermal stress (*Rampelt et al., 2012*) and interacts with protein amyloids in vivo (*Ishihara et al., 2003*; *Wang et al., 2009*; *Olzscha et al., 2011*), raising the possibility that Hsp110 engagement with stress-induced substrates could also promote its activity toward amyloidogenic substrates in vivo.

The spatial engagement of PQC factors, including both chaperones and components of the ubiquitin–proteasome system, is a newly appreciated consequence of their function in vivo. Numerous cytoplasmic foci arise in response to stressors including heat, aging, oxidation, and/or proteasome inhibition. These foci include aggresomes, the insoluble protein deposit (IPOD), the juxtanuclear quality control compartment (JUNQ), StiF-inducible foci (StiF), and Q-bodies, the latter of which form under the mild thermal stress conditions employed in our studies (*Johnston et al., 1998*; *Erjavec et al., 2007*;

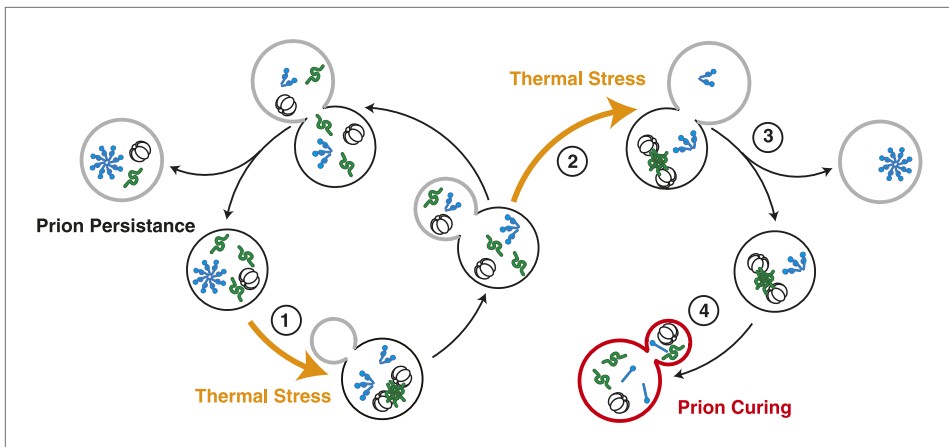

**Figure 8**. Model for Sup35 amyloid resolubilization and curing upon thermal stress. Upon thermal stress, cellular proteins (green) misfold and aggregate, leading to the induction and recruitment of Hsp104 (barrel). If thermal stress occurs in unbudded cells (1), these aggregates are resolved prior to cell division, allowing the partitioning of Hsp104 to both mother (black) and daughter (gray) cells (left). If thermal stress occurs in budded cells (2), heat-induced aggregates persist upon cell division (3), leading to the asymmetric retention of Hsp104 in mother cells. Both heat-induced aggregates (green) and Sup35 amyloid (blue corkscrews) are resolved in cells accumulating high levels of Hsp104, leading to curing (red, 4).

*Kaganovich et al., 2008*; *Liu et al., 2010*; *Specht et al., 2011*; *Malinovska et al., 2012*; *Weisberg et al., 2012*; *Escusa-Toret et al., 2013*; *Wolfe et al., 2013*). While the relationship of each of these foci to one another is currently unclear, they are all defined by the co-localization of misfolded and/or aggregation-prone proteins with PQC factors, some of which can be found in more than one of type of focus. The PQC factors that localize to these foci, such as Hsp104, clearly promote survival under stress (*Sanchez et al., 1992*; *Escusa-Toret et al., 2013*), but whether their localization into cytoplasmic foci specifically altered proteostasis capacity had not been previously established. Our studies indicate that the engagement of Hsp104 with heat-induced misfolded protein aggregates enhances proteostasis capacity by increasing the accumulation of this factor beyond the level attainable by changes in gene expression (*Figure 5*) and thereby permitting the disassembly of existing Sup35 amyloid (*Figures 1,5*).

While our studies indicate that chaperone partitioning imposes a limitation on proteostasis capacity, other aspects of this process may be more relevant to this upper boundary in post-mitotic cells, such as neurons. Indeed, our observations reveal other cell-based limitations beyond chaperone partitioning. For example, in contrast to the proteostasis enhancement we observe following thermal stress in yeast, previous studies have linked the accumulation of protein aggregates to reduced proteostasis capacity in vivo (*Broadley and Hartl, 2009*). In these cases, protein aggregates, including those resulting from oxidative damage with age or proteotoxic stresses, have been linked to reduced replicative lifespan, (*Aguilaniu et al., 2003*; *Hernebring et al., 2006*; *Rujano et al., 2006*; *Erjavec et al., 2007, 2008*; *Tessarz et al., 2009*; *Knorre et al., 2010*; *Liu et al., 2010*; *Unal et al., 2011*; *Zhou et al., 2011*; *Spokoini et al., 2012*) and, the presence of protein amyloids, such as polyglutamine-expanded proteins and other yeast prions, promote the misfolding of metastable proteins, interfere with proteolysis, reduce protein synthesis, inhibit endocytosis, and disrupt prion propagation through the sequestration of chaperones (*Gidalevitz et al., 2006*; *Meriin et al., 2003*; *Kirstein-Miles and Morimoto, 2013*; *Park et al., 2013*; *Yang et al., 2013*; *Yu et al., 2014*). A comparison of these studies with our work suggests that the dynamics of chaperone engagement with distinct substrates, rather than simply their presence, correlates with the impact of these interactions on proteostasis capacity. In the studies resulting in chaperone sequestration, proteotoxicity correlates with imbalances in the system imposed by harsh conditions and/or the unnaturally high expression of amyloidogenic proteins. In contrast, we detect no differences in Hsp104-GFP localization in [*PSI+*] and [*psi−*] strains expressing Sup35 at its endogenous level (*Figure 4A*, 30°C), and Sup35 amyloid can clearly be resolved at these native stoichiometries when the system is elevated to a distinct but accessible

proteostatic niche (*Figure 1D*). However, paralleling the studies in metazoans, [*PSI*⁺] can transition from a benign to a toxic state upon Sup35 overexpression (*Ter-Avanesyan et al., 1993*), a condition that also induces Hsp70 co-localization (*Winkler et al., 2012*). Thus, proteostasis capacity appears to be finely tuned to maintain a natively expressed proteome.

Additional evidence of the importance of this balance can be gleaned by a comparison of chaperone overexpression in different cellular contexts. For example, overexpression of Hsp104 alone, to the same level achieved here through asymmetric retention of this factor (*Table 1* and *Figure 3—figure supplement 1C*), also induces curing (*Chernoff et al., 1995*; *Wegrzyn et al., 2001*). While it has been suggested that Hsp104 overexpression dissolves Sup35 amyloid in vivo, this interpretation is complicated by a lack of temporal resolution and the ability to monitor existing protein (*Park et al., 2014*) and is inconsistent with the increase in the size of SDS-resistant Sup35 aggregates under these conditions (*Figure 3—figure supplement 1D*) (*Kryndushkin et al., 2003*). An alternative model, which is consistent with this biochemical evidence of Hsp104 inhibition, suggests that upon its overexpression, Hsp104 aberrantly and non-productively co-localizes with Sup35 amyloid (*Chernoff et al., 1995*; *Kryndushkin et al., 2003*; *Bagriantsev et al., 2008*; *Winkler et al., 2012*). In contrast, our studies indicate that Hsp104 overexpression within the context of a thermal stress transitions Sup35 amyloid from outside the buffering capacity of the proteostasis network to within its sphere of protection. Notably, Hsp104 co-localization with Sup35 amyloid varies based on its mode of overexpression (i.e. individual vs network up-regulation), again implicating substrate–chaperone dynamics, rather than simply chaperone availability, in proteostasis capacity. Intriguingly, this interplay is distinct for individual PQC factors within the same cell, as thermal stress induces localization of Hsp104, Ssa1, and Sis1 to cytoplasmic foci, but only Hsp104 is asymmetrically retained upon cell division (*Figure 5—figure supplement 1D, E*), suggesting an additional point of proteostasis regulation.

Beyond these cell-based limitations on proteostasis capacity, our studies have deconvoluted the contributions of distinct physical characteristics of amyloid variants to their ability to exceed the PQC buffering capacity in vivo. Intriguingly, we find that [*PSI*⁺]^Weak, the more thermodynamically stable but less efficiently amplified variant of Sup35 amyloid (*Tanaka et al., 2006*), is susceptible to curing at elevated temperature, while the less thermodynamically stable and more efficiently amplified [*PSI*⁺]^Strong variant is not (*Figure 1*), indicating that amyloid amplification rather than stability imposes the primary limitation on amyloid clearance. Consistent with this idea, reducing [*PSI*⁺]^Strong amplification by either expressing a Sup35 mutant or decreasing the expression of wild-type Sup35 promotes curing at elevated temperature (*Figure 2*). Thus, manipulations that have minor effects on the dynamics of existing amyloid are also sufficient to move this alternative protein-folding pathway within the buffering capacity of the proteostasis network.

Together, our observations suggest an alternative to the view that the physical characteristics of amyloid complexes alone preclude their accessibility to the cell's natural defenses against protein misfolding. Rather, the dynamics and balance of the system as a whole, including both protein-based and cell-based contributors, create not only a niche that allows amyloid to arise and persist but also another that promotes amyloid clearance. Our studies, therefore, raise the possibility that the proteostasis limitations that allow the accumulation of chronically misfolded proteins may be distinct in a native context and under conditions of their overexpression. Within this framework, our studies provide a proof-of-principle example to support the idea that proteostasis regulators, which are aimed at transitioning proteostasis landscapes to new thresholds, may be the most effective interventions into amyloidosis (*Lindquist and Kelly, 2011*).

## Materials and methods

### Plasmid and strain construction

All plasmids used in this study were previously reported (*Table 2*) except for SB1013 (pRS306P$_{GPD}$-FFL-mCherry), which contains firefly luciferase as an *Xba1/BamHI* fragment and mCherry as a *BamHI/XhoI* fragment, separated by a three-repeat glycine–serine linker. The ORFs were amplified by PCR using primers 5XbaI firefly/3BamHI firefly and 5BamHIGS3mCherry/3XhoImCherry, respectively (*Table 3*) and confirmed by sequencing. All strains of *Saccharomyces cerevisiae* used in this study are derivatives of 74-D694 (*Table 4*) (*Chernoff et al., 1995*). A WT [*PSI*⁺]^Weak diploid strain (SY945) was generated by mating SY2600 with SLL3252 (*Table 4*). The diploid state was confirmed by sporulation. SY591, a [*PSI*⁺]^Weak strain containing a heterozygous deletion of *HSP104*, was created by transformation of a

**Table 2.** Plasmids

| Name | Description |
|------|-------------|
| SB20 | pRS306-P$_{Sup35}$N(GS)$_3$sGFP(GS)$_3$MC |
| SB503 | pRS304-P$_{GPD}$GST-DsRED-NLS |
| SB630 | pRS306-P$_{GAL}$Hsp104 |
| SB657 | pRS306-P$_{tetO2}$Sup35 |
| SB658 | pRS306-P$_{tetO2}$Sup35(G58D) |
| SB849 | pRS306-P$_{HSE}$GFP |
| SB1013 | pRS306-P$_{GPD}$FFL-mCherry |

*Pvu*I-*Bam*HI fragment of pYABL5 (a kind gift of S. Lindquist) into SY945 and selection on medium lacking leucine. Disruptions were verified by PCR and 2:2 marker segregation upon sporulation and dissection. SY957, a [*PSI*+]$^{Strong}$ diploid strain containing a heterozygous disruption of *SUP35*, was created by transforming a PCR-generated cassette using pFA6a-KanMX4 as a template with primers SD27 and SD28 (*Table 3*) into SLL3071 (*Table 4*). Integration was confirmed by PCR using primers Psup352/PTEFCH and Sup35 3'chk/pFa6 test (*Table 3*). The galactose-inducible *HSP104* strains were made by integrating *Bst*BI-linearized SB630 (*Table 2*) into SY197 (*Table 4*) and selecting transformants on medium lacking uracil. Galactose-inducible expression of Hsp104 was confirmed by western blotting. [*PSI*+]$^{Strong}$ and [*PSI*+]$^{Weak}$ were then cytoduced into this strain from SY1698 and SY1699, respectively, to create SY1748 and SY1749 (*Table 4*), respectively (*Conde and Fink, 1976*). Cytoductants were selected by growth on synthetic medium containing glycerol and lacking uracil and by colony color on YPD. The *HSP104-GFP* [*psi*−] strain (SY2125) was created by transforming a PCR-generated cassette using pFA6a-GFP(S65T)-KanMX6 as a template with primers HSP104-GFP F-A and HSP104-GFP R-A (*Table 3*) into WT [*PSI*+]$^{Strong}$ strains and selection on medium containing 300 µg/ml G418. Integration was confirmed by PCR using primers Hsp104for/GFP-R and pFa6 test/Hsp104 3 flank R (*Table 3*), and expression was confirmed by fluorescence microscopy. These strains were cured of the prion by growth on YPD plates containing 3 mM GdnHCl (*Tuite et al., 1981*). The [*PSI*+]$^{Weak}$ variant (SY2126) was generated by mating SY2125 to a WT [*PSI*+]$^{Weak}$ strain (SLL2600) and sporulation. Tetrads were dissected to recover haploids, and *HSP104-GFP* isolates were verified by G418 resistance, fluorescence microscopy, and quantitative immunoblotting for Hsp104. The heat inducible GFP strain (SY2091) was generated by transformation of a WT [*psi*−] strain (SLL2119) with *Bsu*36I-digested SB849 (*Table 2*). Expression was confirmed by fluorescence microscopy. *SSA1-GFP* (SY2658) and *SIS1-GFP* (SY2447) [*PSI*+]$^{Weak}$ strains were created by transforming PCR-generated cassettes using pFA6a-GFP(S65T)-KanMX6 as a template with primers GFP-GS-Ssa1-F/GFP-Ssa1-R or Sis1-GFP-F GS/Sis1-GFP-R (*Table 3*), respectively, into WT [*PSI*+]$^{Strong}$ strains and selection on medium containing 300 µg/ml G418. Expression was confirmed by fluorescence microscopy and quantitative immunoblotting for Ssa1/2 and Sis1, respectively. These strains were cured of the prion by growth on YPD plates containing 3 mM GdnHCl, mitochondrial loss was induced by growth in 25 µg/ml ethidium bromide, and [*PSI*+]$^{Weak}$ was transferred to them by cytoduction (*Cox, 1965*), using SY1699 as a donor strain. Cytoductants were selected by growth on glycerol medium and 300 µg/ml G418. *SSA1-GFP* and *SIS1-GFP* [*PSI*+]$^{Weak}$ strains containing a nuclear-localized fluorescent reporter protein (DsRed-NLS, SY2659, and SY485, respectively) were generated by transforming SY2658 or SY2447 with *Bsu*36I-digested SB503 (*Table 2*). Expression was confirmed by fluorescence microscopy. The Δ*bni1* [*PSI*+]$^{Weak}$ strain (SY1888) was created by transforming a PCR-generated cassette using pFA6a-KanMX4 as a template with primers AD-BNI1-f and AD-BNI1-r (*Table 3*) into a [*PSI*+]$^{Weak}$ diploid (SY782, a cross between SY2600 and SY86, *Table 4*). Transformants were selected on medium containing 300 µg/ml G418 and verified by PCR using primers AD-BNI1-fseq/PTEFCH and AD-BNI1-rseq/pFa6 test (*Table 3*). The haploid Δ*bni1* [*PSI*+]$^{Weak}$ strain was then generated by sporulation and tetrad dissection and verified by G418 resistance. The *HSP104-GFP* Δ*bni1* [*PSI*+]$^{Weak}$ strain (SY2486) was created by transforming SY2126 with a PCR-generated cassette using pFA6a-hphMX4 as a template with primers AD-BNI1-f and AD-BNI1-r (*Table 3*). Transformants were confirmed by PCR using primers AD-BNI1-fseq/PTEFCH and AD-BNI1-rseq/pFa6 test (*Table 4*) and growth on YPD plates containing 300 µg/ml hygromycin B.

## Growth conditions and phenotypic analysis

Unless otherwise specified, yeast cultures were grown in rich YPD medium supplemented with 0.3 mM adenine. Cultures were maintained at an OD$_{600}$ of less that 0.5 at 30°C for at least 10 doublings to ensure exponential growth. Where indicated, cultures were then transferred to 37°C or 40°C for the specified period. Pretreatment of cultures at 37°C prior to shift to 40°C was for 30 min. To analyze colony color phenotype, aliquots of cultures were diluted in H$_2$O as needed to ensure well-separated single colonies

**Table 3.** Primers

| Name | Sequence |
| --- | --- |
| 5XbaI firefly | 5'-TCTAGAATGGAAGATGCCAAAAACATTAAG-3' |
| 3BamHI firefly | 5'-GGATCCACCTTGAGACTGTGGTTGGAAAC-3' |
| 5BamHI GS3mCherry | 5'-GGATCCGGTAGTGGTAGTGGTAGTATGGTGAGCAAGGG CGAGGAG-3' |
| 3XhoI mCherry | 5'-CTCGAGTTACTTGTACAGCTCGTCCATGCCG-3' |
| SD27 | 5'-ACTTGCTCGGAATAACATCTATATCTGCCCACTAGCAACA CAGCTGAAGCTTCGTACGC-3' |
| SD28 | 5'-GGTATTATTGTGTTTGCATTTACTTATGTTTGCAAGAAATG CATAGGCCACTAGTGGATCTG-3' |
| Psup352 | 5'-GAGATGCTCATCAAGGG-3' |
| PTEFCH | 5'-GCACGTCAAGACTGTCAAGG-3' |
| Sup35 3'chk | 5'-TATTTACGAAGGAGACCCGGAG-3' |
| pFa6 test | 5'-TGCCCAGATGCGAAGTTAAGTG-3' |
| HSP104-GFP F-A | 5'-CGATAATGAGGACAGTATGGAAATTGATGATGACCTA GATCGGATCCCCGGGTTAATTAA-3' |
| Hsp104-GFP R-A | 5'-TATTATATTACTGATTCTTGTTCGAAAGTTTTTAAAAATC GAATTCGAGCTCGTTTAAAC-3' |
| Hsp104for | 5'-GGCACATCCTGATGTTTTGA-3' |
| GFP-R | 5'-CCTTCACCCTCTCCACTGACAG-3' |
| Hsp104 3 flank R | 5'-CCGTATTCTAATAATGGACCAATC-3' |
| GFP-GS-Ssa1-F | 5'-AGCTCCAGAGGCTGAAGGTCCAACCGTTGAAGAAGTTG ATGGTTCTGGTTCTGGTTCTCGGATCCCCGGGTTAATTAA-3' |
| GFP-Ssa1-R | 5'-ACCCAGATCATTAAAAGACATTTTCGTTATTATCAATTGC GAATTCGAGCTCGTTTAAAC-3' |
| Sis1-GFP-F GS | 5'-ACTAAACGACGCTCAAAAACGTGCTATAGATGAAAATTT TGGTTCTGGTTCTGGTTCTCGGATCCCCGGGTTAATTAA-3' |
| Sis1-GFP-R | 5'-ATTTATTTGAGTTTATAATTATATTTGCTTAGGATTACTAG AATTCGAGCTCGTTTAAAC-3' |
| AD-BNI1-f | 5'-ATGTTGAAGAATTCAGGCTCCAAACATTCGAACTCAAAG GCAGCTGAAGCTTCGTACGC-3' |
| AD-BNI1-r | 5'-TTATTTGAAACTTAGCCTGTTACCTGTCCTAGCCTCACCT GCATAGGCCACTAGTGGATCTG-3' |
| AD-BNI1-fseq | 5'-GACATCGGTTAGAGGAAG-3' |
| AD-BNI1-rseq | 5'-CACTGTGCTTGTCACTTA-3' |

upon plating to solid YPD medium. After growth at 30°C, each colony was scored based on colony color phenotype: fully cured (completely red, [*psi⁻*]), sectored (part red and part white), or [*PSI⁺*] (completely white). Unless otherwise indicated, fully cured and sectored colonies were combined in the 'cured' category. For all colony counting assays, at least 150 colonies were counted for each experimental condition/timepoint. For the galactose-inducible Hsp104 experiments, cells were grown in rich YP medium containing 3% raffinose supplemented with 3% galactose during induction. α-factor and nocodazole arrests were performed in YPD liquid medium containing final concentrations of 5 µg/ml α-factor or 15 µg/ml nocodazole, respectively, for ~2 hr. Following confirmation of arrest based on cell morphology by bright-field microscopy, cultures were washed three times with medium containing 1 mM DMSO followed by one wash in YPD before resuspension for indicated manipulation. GdnHCl treatment was performed at 3 mM final concentration in liquid YPD, and for experiments involving recovery, cultures were washed three times with medium before resuspension in YPD for indicated manipulation.

## Protein analysis

SDS-PAGE and quantitative immunoblotting were performed as previously described (*Pezza et al., 2009*). Anti-Ssa1/2 rabbit serum was provided by E. Craig (U Wisconsin—Madison), and anti-Sis1 rabbit

**Table 4.** Yeast strains

| Strain | Genotype | Plasmids integrated | Reference | Figure |
|---|---|---|---|---|
| SLL2119 | MATa [psi⁻] ade1-14 his3Δ200 trp1-289 ura3-52 leu2-3, 112 | - | *Chernoff et al., 1995* | 1c, 3Sf |
| SLL2600 | MATa [PSI⁺]$^{Weak}$ ade1-14 his3Δ200 trp1-289 ura3-52 leu2-3, 112 | - | *Derkatch et al., 1996* | 1, 2a, 3, 5ce, 6abcfg, 1S, 3Sabg, 4Sabcd, 5Sfg, 6Sa |
| SLL2606 | MATa [PSI⁺]$^{Strong}$ ade1-14 his3Δ200 trp1-289 ura3-52 leu2-3, 112 | - | *Chernoff et al., 1995* | 1ac, 3Sf |
| SLL3071 | MATa/α [PSI⁺]$^{Strong}$ ade1-14/ade1-14 his3Δ200/his3Δ200 trp1-289/ trp1-289 ura3-52/ura3-52 leu2-3112/ leu2-3112 | - | *DiSalvo et al., 2011* | 2d |
| SLL3252 | MATa [psi⁻] ade1-14 his3Δ200 trp1-289 ura3-52 leu2-3, 112 | - | *Chernoff et al., 1995* | 'Materials and methods' |
| SY86 | MATα [psi⁻] ade1-14 his3Δ200 trp1-289 ura3-52 leu2-3, 112 sup35::N(GS)₃GFP(GS)₃MC | SB20 | *Derdowski et al., 2010* | 'Materials and methods' |
| SY197 | MATa [psi⁻] ade1-14 his3-11,-15 trp1-1 ura3-1 leu2-3112 can1-100 | - | J Weissman (YJW513) | 'Materials and methods' |
| SY591 | MATa/α [PSI⁺]$^{Weak}$ ade1-14/ade1-14 his3Δ200/his3Δ200 TRP/ trp1-289 ura3-52/ura3-52 leu2-3112/ leu2-3112 HSP104/hsp104::LEU2 | - | This study | 2b |
| SY782 | MATa/α [PSI⁺]$^{Weak}$ ade1-14/ade1-14 his3Δ200/his3Δ200 trp1-289/ trp1-289 ura3-52/ura3-52 leu2-3112/ leu2-3112 SUP35/sup35::N(GS)₃GFP(GS)₃MC | - | This study | 'Materials and methods' |
| SY945 | MATa/α [PSI⁺]$^{Weak}$ ade1-14/ade1-14 his3Δ200/his3Δ200 trp1-289/ trp1-289 ura3-52/ura3-52 leu2-3112/ leu2-3112 | - | This study | 2b |
| SY957 | MATa/α [PSI⁺]$^{Strong}$ ade1-14/ade1-14 his3Δ200/his3Δ200 trp1-289/ trp1-289 ura3-52/ura3-52 leu2-3112/ leu2-3112 SUP35/sup35::KANMX4 | - | This study | 2d |
| SY1646 | MATa/α [PSI⁺]$^{Strong}$ ade1-14/ade1-14 his3Δ200/his3Δ200 trp1-289/ trp1-289 ura3-52/ura3-52::URA3::P$_{tetO2}$SUP35 leu2-3112/ leu2-3112 SUP35/ sup35::KANMX4 | SB657 | *DiSalvo et al., 2011* | 2c |
| SY1648 | MATa/α [PSI⁺]$^{Strong}$ ade1-14/ade1-14 his3Δ200/his3Δ200 trp1-289/ trp1-289 ura3-52/ura3-52::URA3::P$_{tetO2}$ SUP35(G58D) leu2-3112/ leu2-3112 SUP35/sup35::KANMX4 | SB658 | *DiSalvo et al., 2011* | 2c |
| SY1698 | MATα [PSI⁺]$^{Strong}$ ade1-14 his3Δ200 ura3-52 leu2-3 kar1-d15 ConR CyhR | - | This study | 'Materials and methods' |
| SY1699 | MATα [PSI⁺]$^{Weak}$ ade1-14 his3Δ200 ura3-52 leu2-3 kar1-d15 ConR CyhR | - | This study | 'Materials and methods' |
| SY1748 | MATa [PSI⁺]$^{Strong}$ ade1-14 his3-11,-15 trp1-1 ura3-1::URA3::P$_{GAL}$HSP104 leu2-3112 can1-100 | SB630 | This study | 3Se |
| SY1749 | MATa [PSI⁺]$^{Weak}$ ade1-14 his3-11,-15 trp1-1 ura3-1::URA3::P$_{GAL}$HSP104 leu2-3112 can1-100 | SB630 | This study | 3Scde |
| SY1888 | MATa [PSI⁺]$^{Weak}$ ade1-14 his3Δ200 trp1-289 ura3-52 leu2-3, 112 Δbni1::KANMX4 | - | This study | 5c, 5Sfg |
| SY2091 | MATa [psi⁻] ade1-14 his3Δ200 trp1-289 ura3-52::URA::P$_{HSE}$GFP leu2-3, 112 | SB849 | This study | 5Sa |

*Table 4. Continued on next page*

Table 4. Continued

| Strain | Genotype | Plasmids integrated | Reference | Figure |
|--------|----------|---------------------|-----------|--------|
| SY2125 | MATα [psi⁻] ade1-14 his3Δ200 trp1-289 ura3-52 leu2-3112 HSP104GFP::KANMX6 | - | This study | 4b |
| SY2126 | MATa [PSI⁺]$^{Weak}$ ade1-14 his3Δ200 trp1-289 ura3-52 leu2-3112 HSP104GFP::KANMX6 | - | This study | 4, 5abdfg, 6de, 7, 4Sabcdfg, 6Sb |
| SY2447 | MATa [PSI⁺]$^{Weak}$ ade1-14 his3Δ200 trp1-289 ura3-52 leu2-3112 SIS1GFP::KANMX6 | - | This study | 5Se |
| SY2485 | MATa [PSI⁺]$^{Weak}$ ade1-14 his3Δ200 trp1-289::TRP::P$_{GPD}$GST-DsRed-NLS ura3-52 leu2-3112 SIS1GFP::KANMX6 | SB503 | This study | 5Sc |
| SY2486 | MATa [PSI⁺]$^{Weak}$ ade1-14 his3Δ200 trp1-289 ura3-52 leu2-3, 112 HSP104GFP::KANMX6 Δbni1::hphMX4 | - | This study | 5b |
| SY2658 | MATa [PSI⁺]$^{Weak}$ ade1-14 his3Δ200 trp1-289 ura3-52 leu2-3112 SSA1GFP::KANMX6 | - | This study | 5Sd |
| SY2659 | MATa [PSI⁺]$^{Weak}$ ade1-14 his3Δ200 trp1-289::TRP::P$_{GPD}$GST-DsRed-NLS ura3-52 leu2-3112 SSA1GFP::KANMX6 | SB503 | This study | 5Sb |
| SY2802 | MATa [PSI⁺]$^{Weak}$ ade1-14 his3Δ200 trp1-289 ura3-52::URA::P$_{GPD}$ Firefly-mCherry leu2-3112 HSP104GFP::KANMX6 | SB1013 | This study | 4Se |

serum was provided by M. Tuite (U Kent, Canterbury, UK). Semi-native agarose gel electrophoresis (SDD-AGE) was performed as previously described (*Kryndushkin et al., 2003*). The cycloheximide SDS-sensitivity assay was performed as previously described (*DiSalvo et al., 2011*) with the following modifications: 1) cultures were treated at the various experimental temperatures for 30 min prior to the addition of cycloheximide to allow for the induction of chaperone proteins, and 2) after cycloheximide treatment, cultures were incubated with shaking at 30°C for 2 hr before lysis and analysis. For the aggregation analysis, native lysates were prepared as described previously (*Kryndushkin et al., 2003*). Lysates were pre-cleared for 1 min at 500×*g* and total protein content was quantified using the BioRad Bradford assay in triplicate. Lysates were subjected to 15,000×g centrifugation for 15 min, and pellets were washed with 10 mM sodium phosphate buffer (pH7.5) containing 2% NP-40 before being resuspended in 10 mM sodium phosphate buffer (pH7.5) and quantified again in triplicate using the Bradford assay. For the Hsp104 immunocapture, native lysates were prepared at 4°C in IP buffer (50 mM HEPES–NaOH (pH 7.5), 150 mM NaCl, 10 mM MgCl$_2$, 1 mM EDTA, 1% NP-40, 0.25% Na-deoxycholate, and protease inhibitors (2 mM PMSF, 5 µg/ml pepstatin, complete protease inhibitor tablets (Pierce, Rockford, IL), protease inhibitor cocktail (Sigma-Aldrich, St. Louis, MO)). Lysates were pre-cleared for 1 min at 500×*g*, and then incubated for 1 hr with Protein G magnetic beads (NEB, Ipswich, MA) with nutation. Immunocapture was performed using Protein G magnetic beads and anti-GFP mouse monoclonal antibody (Roche, Switzerland). Beads were washed 4× with IP buffer and 1× with 50 mM HEPES–NaOH (pH 7.5), and protein was eluted by boiling in SDS sample buffer. Co-captured proteins were resolved by SDS-PAGE and analyzed by gel staining with Flamingo (Bio-Rad, Hercules, CA) and fluorescent scanning on a Typhoon imager (GE Lifesciences, Marlborough, MA) according to the manufacturer's instructions or by western blot for GFP.

## Imaging and microfluidics

Static images were obtained on a Zeiss Axio Imager M2 fluorescence light microscope equipped with a 100× objective. Confocal images were obtained on a Zeiss LSM 510 Meta confocal microscope using a 100× objective. Microfluidics experiments were performed on a Zeiss Axio Observer Z1 using

a CellAsics microfluidics plate with temperature controls and media flow of 2 psi on a Y0C4 yeast perfusion plate (channel size 3.5–5 µm). Imaging was performed in complete minimal medium supplemented with 2% glucose and 2.5 mM adenine. Fluorescence intensity was analyzed using the Zen software package (Zeiss, Germany).

## Flow Cytometry

Flow cytometry and cell sorting was performed on a BD FACSAria fluorescence-activated cell sorter using a 488 nm laser and a FITC-A filter to measure GFP fluorescence intensity in single cells. Data were obtained at least in triplicate with representative spectra shown. Data were analyzed using the FlowJo software package (TreeStar Inc., Ashland, OR).

## Acknowledgements

We thank Jeff Laney, members of the Serio and Laney laboratories, and Ulrich Hartl for helpful discussions and comments on the manuscript, and E Craig and M Tuite for antisera. This work was funded by awards from the NIH to TRS (R01 GM069802001), CLK (F31 AG034754), and CRL (F31 GM099383).

# Additional information

## Funding

| Funder | Grant reference number | Author |
|---|---|---|
| National Institute of General Medical Sciences | R01 GM069802001 | Tricia R Serio |
| National Institutes of Health | F31 AG034754 | Courtney L Klaips |
| National Institute of General Medical Sciences | F31 GM099383 | Christine R Langlois |

The funders had no role in study design, data collection and interpretation, or the decision to submit the work for publication.

## Author contributions

CLK, Conception and design, Acquisition of data, Analysis and interpretation of data, Drafting or revising the article; MLH, CRL, Acquisition of data, Analysis and interpretation of data; TRS, Conception and design, Analysis and interpretation of data, Drafting or revising the article

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
