## [Decision Letter]

Thank you for sending your work entitled “Spatial Quality Control Bypasses Cell-Based Limitations on Proteostasis To Promote Prion Curing” for consideration at *eLife.* Your article has been favorably evaluated by Randy Schekman (Senior editor), Jeffery W Kelly (Reviewing editor), and 3 reviewers.

The Reviewing editor and the reviewers discussed their comments before we reached this decision, and the Reviewing editor has assembled the following comments to help you prepare a revised submission.

Serio and co-workers have submitted a well written and reasonably convincing manuscript demonstrating that spatial temporal retention of Hsp104 by [*PSI*^*+*^]^Weak^ cells induces curing through the disassembly of existing Sup35 amyloid by Hsp104. Serio et al. showed that Hsp104 must engage heat-induced protein aggregates for a period that exceeds the time to the next cell division. As a result, Hsp104 is asymmetrically localized to the cells that experienced the thermal stress (Figures 4 and 7), and this increase in chaperone accumulation promotes [*PSI*^*+*^]^Weak^ curing in the same cells (Figures 4 and 7). This study provides evidence that the engagement of Hsp104 with heat-induced misfolded Sup35 aggregates enhances proteostasis capacity by increasing the accumulation of this factor, perhaps beyond the level attainable by changes in gene expression (Figure 4), thereby enabling the disassembly of existing Sup35 amyloid (Figures 1 and 4), restoring homeostasis to the mother cell. While the spatial engagement of proteostasis network components is not new, there are enough novel attributes to this work to render it an interesting *eLife* paper.

Please carefully consider the three thoughtful reviews pasted below when revising your manuscript. The points common to at least two reviews include:

1) It is important for the model presented in this work that Hsp104 interacts specifically with aggregated proteins, and that this interaction is dependent on the enzymatic activity of Hsp104. Is it possible to demonstrate this specific interaction, e.g. by co-immunoprecipitation? How does the sequestration of Hsp104 by heat-induced aggregates affect its activity? Is it solely the consequence of achieving high local concentrations?

2) A broader context here would be useful as a number of laboratories have made observations on Q-bodies (22), iPOD, JUNQ (37), aggresomes (34), and spatial sequestration and symmetrical inheritance in yeast (76) have demonstrated that aggregates have spatial restrictions and numerous papers have shown that multiple chaperones are associated with these aggregate structures using a wide range of proteomic and co-localization methods, often with substantial consequences on cellular activity. Some of these would seem to have opposing outcomes, which then poses questions how the mother-daughter cell relationship of yeast relates to metazoan cell division, and broader relevance to neurons being post-mitotic. Integrating these observations into this work without dramatically expanding the Discussion would be useful.

3) The use of Sup35 as a client and the yeast cellular environment while highly relevant on its own, may not offer broadly relevant concepts than can be extrapolated to metazoans. For example, Hsp104 as a disaggregase activity is restricted to prokaryotes, plants, and yeast whereas a corresponding class of activity in metazoans appears to be a reconfiguration of the Hsp70 and J-domain apparatus by Hsp110, leading to questions whether these concepts on Hsp104 are specific to yeast (which is still interesting) or can help us to understand the larger questions posed by the authors on the upper limit of the cellular environment. Some brief comments addressing this issue will allow the reader to comprehend the bigger picture.

*Reviewer #1*:

Protein aggregation is linked to many degenerative diseases, either through loss of function of the aggregated protein, or through a novel gain of toxic function of the aggregate. It is therefore of high importance to identify pathways that can prevent or reverse protein aggregation. The submitted manuscript investigates the relationship between protein aggregation and the aggregate-remodeling factor Hsp104.

The authors report the seemingly paradoxical observation that incubation of cells at an elevated temperature of 40^°^C strongly reduces aggregation of the model substrate Sup35 in its prion form [*PSI*^*+*^]^Weak^. Heat shock treatment reduces the size of the aggregates and increases the amount of soluble Sup35 after a recovery period. This process is dependent on the activity of Hsp104. The authors convincingly demonstrate that heat-induced aggregates of proteins other than Sup35 asymmetrically retain Hsp104 during cell division, in a process that is dependent of the enzymatic activity of Hsp104, and thereby increase Hsp104 concentration in mother cells to a level that allows Sup35 disaggregation and prion curing during recovery from heat stress.

The concept that the (temporary) sequestration of important proteostasis factors into an aggregate can be beneficial instead of detrimental is novel and exciting, and provides insight into an unanticipated role of asymmetric cell division in protein aggregate management. The paper is clearly written and the main conclusions are generally well supported by data.

There are however a few points the authors should address:

1) It is important for the model presented in this work that Hsp104 interacts specifically with aggregated proteins, and that this interaction is dependent on the enzymatic activity of Hsp104. Is it possible to demonstrate this specific interaction, e.g. by co-immunoprecipitation?

2) The use of nocodazole in Figure 5 seems problematic as it may influence several other pathways that are involved in the formation and degradation of aggregates. Since the cell cycle analysis is not contributing critically to the overall model, it might be better to remove this section, or move it to the supplement.

3) Some of the colors in Figure 4 are hard to distinguish (especially dark blue and black in 4d) and are not explained in the panels.

*Reviewer #2*:

Klaips et. al. (Serio) have explored the mechanism of curing of Sup35 prion aggregates upon exposure of yeast to transient thermal stress to test a hypothesis that the cellular environment sets limits on protein quality control (PQC). The authors show that exposure of yeast (S. cerevisiae) at specific stages of the cell cycle to short periods of thermal stress leads to the disassembly of Sup35 amyloid, and propose that this is due to the spatial sequestration of Hsp104 in the mother cell rather than the result of overall elevated levels of Hsp104 or other chaperones. The sequestration of Hsp104 is correlated with the accumulation of heat-induced, non-prion aggregates upon thermal stress and the data suggest that the disassembly of Sup35 is more a function of amplification efficiency than thermodynamic stability. FACS experiments coupled with the use of two different cell-cycle inhibitors demonstrated that prion curing was highly dependent upon the cell-cycle stage.

Overall, the experiments are well done, the results are intriguing and the paper is nicely presented. Some of the conclusions are correlative rather than causal, thus limiting the overall enthusiasm. Also, the use of Sup35 as a probe and the yeast cellular environment while highly relevant on its own, may not offer broadly relevant concepts than can be extrapolated to metazoans. For example, Hsp104 as a disaggregase activity is restricted to prokaryotes, plants, and yeast whereas a corresponding class of activity in metazoans appears to be a reconfiguration of the Hsp70 and J-domain apparatus by Hsp110, leading to questions whether these concepts on Hsp104 are specific to yeast (which is still interesting) or can help us to understand the larger questions posed by the authors on the upper limit of the cellular environment. How does the sequestration of Hsp104 by heat-induced aggregates affect its activity? Is it solely the consequence of achieving high local concentrations? An argument is made that it is the asymmetrical partitioning of Hsp104 by its association with thermal aggregates that leads to a sufficiently high concentration of Hsp104 to resolve Sup35 amyloids. What is this concentration, what is the nature and basis of the interaction? Will elevated levels of Hsp104 achieved using conditional systems or shield-based metastable DHFR achieve the same outcome? Finally, the effects of different temperature conditions are intriguing. What is the nature of the biophysical transformation of Sup35 in 30C cells compared to those exposed to 40C beyond that rather vague terminology of shift from SDS-resistant to SDS-sensitive? What is special about 40C? Is this a useful tool to understand more completely the cellular environment?

A broader context here would be useful as a number of laboratories have made observations on Q-bodies (22), iPOD, JUNQ (37), aggresomes (34), and spatial sequestration and symmetrical inheritance in yeast (76) have demonstrated that aggregates have spatial restrictions and numerous papers have shown that multiple chaperones are associated with these aggregate structures using a wide range of proteomic and co-localization methods, often with substantial consequences on cellular activity. Some of these would seem to have opposing outcomes, which then poses questions how the mother-daughter cell relationship of yeast relates to metazoan cell division, and broader relevance to neurons being post-mitotic.

Additional comments:

1) Evidence for involvement of Hsp104 in Sup35 amyloid disassembly comes both from the literature and new data presented here, that (i) the GdnHCl curing of [*PSI*^*+*^]^Weak^ and a ∼50% decrease in curing efficiency was observed in a heterozygous deletion of Hsp104 (Figure 2), (ii) Hsp104 was asymmetric localized upon thermal stress (Figures 4 and 6) and (iii) no significant changes were observed for Hsp104, Ssa1 and Sis1 chaperone expression levels upon heat stress (Figure 3). This establishes necessity, but not sufficiency and therefore does not rule out other 'factor(s)' induced upon heat stress that could contribute to amyloid disassembly.

These arguments can be strengthened by: (a) demonstrating whether the expression of other chaperone candidates (ie., other Hsp70 and Hsp40 isoforms) are affected by these thermal stress conditions. For example, there could be small changes in the expression of a number of chaperones that affects the cellular environment and enables amyloid resolubilization. Examples of candidates to test include chaperones examined in studies by [7]; [75]; [17] and [94], and (b) showing the over-expression data, currently not shown (p.18) that establishes that the o/e of Hsp104 alone, to the same level achieved in this study through its asymmetric retention, suffices to induce curing.

2) The authors' evidence for the formation of heat-induced, non-prion aggregates that accumulate upon heat stress and sequester Hsp104 in the mother cell rests on SDD-AGE (Figure 1), differential centrifugation (Figure 3) and that formation of aggregates correlates with other Hsp104 specific observations.

The protocol used for differential centrifugation is not sufficiently well described. It appears that the aggregates were quantified using a Bradford assay after pelleting in a tabletop centrifuge and a mild detergent wash. If the aggregates were not denatured before quantification, the presence of large particles may cause artifacts while making colorimetric measurements in a spectrophotometer. Since the quantification of these aggregates is an important aspect of this story, the authors should quantify aggregate formation with more vigor, using alternative procedures such as sucrose-density gradients and ultra-centrifugation to separate aggregates from other cellular debris. More central, the differential accumulation of aggregates in mother vs. daughter cells should be quantified and compared to the values on the differential retention of Hsp104.

3) An important conclusion is that the heat-induced aggregates directly retain Hsp104, but there is no evidence for the direct interaction between these two factors, other than strong correlation. This would be greatly strengthened by an experiment that isolates these aggregates (see point 2 above) and demonstrates the presence of Hsp104 bound to these aggregates, using a digest and mass spectrometric quantification, for example. Alternatively, the authors could perform a co-IP experiment if they had more information on one of the proteins that is a constituent of these heat-induced aggregates.

4) Have the authors ruled out an effect on protein synthesis? Heat shock, in particular, is well known to inhibit protein synthesis, which could affect the flux, and shift equilibria in the proper cellular environment towards dissociation?

5) A point is regarding the relevance of the conclusions the authors make on the existence of cell-based limitations that preclude amyloid resolubilization in vivo to disease conditions. Since it is clear from their data that the cell-cycle stage is an important determinant that affects the success of resolubilization in vivo, this would not influence our understanding of the PQC in non-dividing neuronal cells. The authors should place their results in the broader context of protein misfolding diseases in the discussion.

6) Another point in the discussion on spatial engagement of PQC factors and effects on proteostasis capacity is a recent observation (99) on aggregate-associated sequestration of Hsc70 leading to down regulation of clathrin-mediated endocytosis. Does this suggest that the interactions of chaperones with aggregates could have different outcomes?

Reviewer #3:

In this study, Klaips et al. perform an elegant set of studies, which demonstrate that Hsp104 dissolves Sup35 prions that encode weak [PSI+] at elevated temperatures in yeast. Interestingly, this curing process depends upon an unprecedented spatiotemporal mechanism. Thus, asymmetric Hsp104 retention by heat induced, non-prion aggregates in late cell cycle-stage cells, leads to high enough Hsp104 levels to promote dissolution of Sup35 prions encoding weak [PSI+]. Upon recognizing this potential mechanism, the authors then cleverly tweak the system to enable curing of Sup35 prions that encode strong [PSI+]. Although Sup35 prion dissolution by Hsp104 is not very surprising per se, the spatiotemporal mechanism revealed by these studies is entirely novel and should interest a broad audience. Indeed, this paper will raise awareness that spatiotemporal mechanisms can have a profound impact on proteostasis and amyloid resolution. In my view, the study is high quality and is carefully conducted and controlled, and I do not have any issues with the experiments or data as presented. A limitation is that several conclusions are based upon correlative evidence, but it has already been shown in vitro that Hsp104 can rapidly dissolve Sup35 prions. My only minor objections concern several statements in the Introduction that are incorrect or misleading:

1) 'but a direct demonstration of amyloid resolubilization in vivo has yet to be reported in any system'. Serio and colleagues have themselves already convincingly demonstrated amyloid resolublization by Hsp104 in vivo in their 2011 NSMB paper (15). [61] also provide compelling evidence that Hsp104 promotes amyloid solubilization in vivo. Moreover, there are numerous examples of amyloid clearance in conditional animal models of neurodegenerative disease models (e.g. Yamamoto et al. Cell 2000; Lim et al., J. Neuorosci. 2011). Hence, this statement is incorrect and misleading.

2) 'Rather, the effects of chaperone overexpression have been shown in some cases to be independent of their catalytic function (5, 6, 33, 61)'. This statement is a little misleading as in several cases the effect has been shown to depend on catalytic function (e.g. [12] PLoS Genet.)

3) 'Extracts from *C. elegans* and mammalian tissues and cell lines similarly promote amyloid solubilization (Cohen et al., 53 2006, Murray et al., 2010).' This statement should also be revised since Murray et al. subsequently published (Protein Sci. 2013 Nov;22(11):1531-41.) that 'our interpretation of the kinetic fibril disaggregation assay data previously reported in Bieschke et al., Protein Sci 2009;18:2231-2241 and Murray et al., Protein Sci 2010;19:836-846 is invalid when used as evidence for a disaggregase activity.'

---

## [Author Response]

*1) It is important for the model presented in this work that Hsp104 interacts specifically with aggregated proteins, and that this interaction is dependent on the enzymatic activity of Hsp104. Is it possible to demonstrate this specific interaction, e.g. by co-immunoprecipitation*?

We agree with the reviewers that the interaction of Hsp104 with heat-induced aggregates is an essential component of our model. To address this point, we have added two new pieces of data to the manuscript. First, we demonstrate that Hsp104 co-localizes with a model substrate (firefly luciferase-mCherry), which denatures upon thermal stress and is reactivated in an Hsp104-dependent manner (Figure 4—figure supplement 1; Parsell et al. 1994). Second, we demonstrate by immunocapture of Hsp104-GFP, SDS-PAGE and general protein staining, that Hsp104-GFP associates with a larger number of proteins upon thermal stress (Figure 4). Importantly, the number of co-captured proteins increases with the severity of the stress (i.e. 40°C > 37°C) and decreases with GdnHCl treatment (Figure 4), which blocks Hsp104 activity. Thus, the engagement of Hsp104-GFP with heat-induced substrates parallels the efficiency of [*PSI*^*+*^]^Weak^ curing, which is increased at 40°C relative to 37°C (Figure 1) and is inhibited by treatment with GdnHCl (Figure 2), providing additional support for our model of prion curing.

*How does the sequestration of Hsp104 by heat-induced aggregates affect its activity? Is it solely the consequence of achieving high local concentrations*?

Our studies together indicate that both the high local concentration of Hsp104 and its activity are required for [*PSI*^*+*^]^Weak^ curing. To summarize here, curing correlates only with the highest levels of Hsp104 accumulation (37°C → 40°C and 40°C; Figure 1 and Figure 5) and is blocked by treatments that reduce Hsp104 asymmetric accumulation (GdnHCl treatment [Figures 2 and 5] and disruption of *BNI1* [Figure 5]). In addition to these observations, we now include an additional experiment, in which cells are treated with GdnHCl after the thermal stress (Figure 5—figure supplement 1 and Figure 5). Under these conditions, Hsp104-GFP still localizes to heat-induced aggregates (Figure 4—figure supplement 1) and is asymmetrically retained (Figure 5), but [*PSI*^*+*^]^Weak^ curing is reduced (Figure 4—figure supplement 1), indicating that both Hsp104-GFP localization and activity are required for curing.

Prolonged overexpression of Hsp104 from a galactose-inducible promoter alone has been previously reported to induce prion curing of both [*PSI*^*+*^]^Strong^ (7) and [*PSI*^*+*^]^Weak^ (92). We have also previously shown that lower-level overexpression (∼5-fold) is insufficient to cure [*PSI*^*+*^]^Strong^ (15). Using short induction of a galactose-inducible Hsp104 construct that transiently raises Hsp104 accumulation to the level observed with thermal stress (Figure 3—figure supplement 1), we now demonstrate that [*PSI*^*+*^]^Weak^ curing does occur, but at a lower frequency than with thermal stress (Figure 1 and Figure 3—figure supplement 1). Studies from the Ter-Avanesyan lab revealed that Sup35 SDS-resistant aggregates isolated from a [*PSI*^*+*^]^Strong^ strain increase in size upon Hsp104 overexpression (45). We have confirmed this result and further demonstrated that the same holds true for SDS-resistant aggregates isolated from a [*PSI*^*+*^]^Weak^ strain (Figure 3—figure supplement 1). This observation is in contrast to our analysis of Sup35 SDS-resistant aggregates following thermal stress. Under thermal stress, both [*PSI*^*+*^]^Strong^ and [*PSI*^*+*^]^Weak^ aggregates decrease in size (Figure 1), and in the latter case, are disassembled to release soluble Sup35 (Figure 1). As we address in the Discussion, these observations together indicate that the mechanisms of curing by specific overexpression of Hsp104 and by thermal stress are distinct.

*2) A broader context here would be useful as a number of laboratories have made observations on Q-bodies (*[22]*), iPOD, JUNQ (*[37]*), aggresomes (*[34]*), and spatial sequestration and symmetrical inheritance in yeast (*[76]*) have demonstrated that aggregates have spatial restrictions and numerous papers have shown that multiple chaperones are associated with these aggregate structures using a wide range of proteomic and co-localization methods, often with substantial consequences on cellular activity. Some of these would seem to have opposing outcomes, which then poses questions how the mother-daughter cell relationship of yeast relates to metazoan cell division, and broader relevance to neurons being post-mitotic. Integrating these observations into this work without dramatically expanding the Discussion would be useful*.

We thank the reviewers for these helpful suggestions. We have reworked the Discussion to more clearly link our studies to the literature available on spatial quality control and to address the distinctions between our observations of enhanced proteostasis capacity and those in the literature demonstrating reduced proteostasis capacity. We believe that these distinct outcomes arise from imbalance between chaperones and substrates created by significant perturbations of the cellular environment beyond what can be achieved naturally.

While the limitations on proteostasis capacity imposed by chaperone partitioning are clearly not applicable to post-mitotic neurons, we believe that the issues raised by a comparison of our work with those previous studies highlight the importance of system balance, particularly the role of chaperone-substrate dynamics in determining proteostasis capacity. These latter points are likely to be independent of cell-type and are clarified in the revised Discussion.

*3) The use of Sup35 as a client and the yeast cellular environment while highly relevant on its own, may not offer broadly relevant concepts than can be extrapolated to metazoans. For example, Hsp104 as a disaggregase activity is restricted to prokaryotes, plants, and yeast whereas a corresponding class of activity in metazoans appears to be a reconfiguration of the Hsp70 and J-domain apparatus by Hsp110, leading to questions whether these concepts on Hsp104 are specific to yeast (which is still interesting) or can help us to understand the larger questions posed by the authors on the upper limit of the cellular environment. Some brief comments addressing this issue will allow the reader to comprehend the bigger picture*.

The reviewer raises an important point that we failed to address in the original submission. We have now included a paragraph on mammalian disaggregase activity in the Discussion, including the similarities and distinctions with the yeast system.

Reviewer #1:

*1) It is important for the model presented in this work that Hsp104 interacts specifically with aggregated proteins, and that this interaction is dependent on the enzymatic activity of Hsp104*. *Is it possible to demonstrate this specific interaction, e.g. by co-immunoprecipitation?*

Please see the response to point 1 under “Response to points common to at least two reviews” above.

*2) The use of nocodazole in*
Figure 5
*seems problematic as it may influence several other pathways that are involved in the formation and degradation of aggregates. Since the cell cycle analysis is not contributing critically to the overall model, it might be better to remove this section, or move it to the supplement*.

We were also concerned about off-pathway effects of nocodazole treatment, which prompted us to assess both Hsp104 expression and localization in the presence of this treatment (Figure 6—figure supplement 1), and to further test the suggestions raised by these arrest experiments using single-cell analyses in asynchronous cultures (Figure 7). We found no deviations from unarrested cells in either case. We believe that the cell-cycle dependence of the curing effect (Figure 6) is an important aspect of the model because it identifies the importance of heat-induced aggregate resolution relative to cell division as a key component of [*PSI*^*+*^]^Weak^ curing. Both reviewer 2 and reviewer 3 concur with the importance of this point in their reviews, and we have therefore retained the figure in the main figures of the revised manuscript.

*3) Some of the colors in*
Figure 4
*are hard to distinguish (especially dark blue and black in 4d) and are not explained in the panels*.

We have changed the colors in panel Figure 4 (Figure 5 in the revised manuscript) to green to increase contrast and have updated the figure legend to reflect this change.

Reviewer #2:

*Some of the conclusions are correlative rather than causal, thus limiting the overall enthusiasm*.

We have addressed the specific comments of this reviewer below.

*Also, the use of Sup35 as a probe and the yeast cellular environment while highly relevant on its own, may not offer broadly relevant concepts than can be extrapolated to metazoans. For example, Hsp104 as a disaggregase activity is restricted to prokaryotes, plants, and yeast whereas a corresponding class of activity in metazoans appears to be a reconfiguration of the Hsp70 and J-domain apparatus by Hsp110, leading to questions whether these concepts on Hsp104 are specific to yeast (which is still interesting) or can help us to understand the larger questions posed by the authors on the upper limit of the cellular environment*.

Please see the response to point 4 under “Response to points common to at least two reviews” above.

*How does the sequestration of Hsp104 by heat-induced aggregates affect its activity*? *Is it solely the consequence of achieving high local concentrations?*

Please see the response to point 2 under “Response to points common to at least two reviews” above.

*An argument is made that it is the asymmetrical partitioning of Hsp104 by its association with thermal aggregates that leads to a sufficiently high concentration of Hsp104 to resolve Sup35 amyloids. What is this concentration, what is the nature and basis of the interaction*?

On the basis of our microfluidic analyses, curing of [*PSI*^*+*^]^Weak^ occurs when Hsp104 levels are increased ∼3.5-fold by thermal stress, relative to 30°C (Table 1). The interaction of Hsp104 with heat-induced substrates, as assessed by localization (Figure 4—figure supplement 1) and immunocapture (Figure 4), requires its enzymatic activity (i.e. is blocked by GdnHCl treatment). These points have been clarified in the text with the addition of new data as detailed above.

*Will elevated levels of Hsp104 achieved using conditional systems or shield-based metastable DHFR achieve the same outcome*?

Please see response to point 2 under “Response to points common to at least two reviews” above for a response to the question on specific elevation of Hsp104 levels.

For model substrates, we have not attempted this precise experiment; however, in our other recently published work (29), we note that the appearance of misfolded proteins in a [*PSI*^*+*^] yeast strain disrupted for the N-terminal acetyltransferase NatA leads to a decrease in the size of SDS-resistant Sup35 aggregates, but this effect is not sufficient to induce prion curing. Under these conditions, Hsp104 expression is elevated ∼3-fold, but we see no evidence of asymmetric localization by microscopy presumably due to the continued misfolding of newly-made proteins in both mother and daughter cells. Thus, both elevation of Hsp104 and its asymmetric localization are required for [*PSI*^*+*^]^Weak^ curing, as noted above in point 2 under “Response to points common to at least two reviews.”

*Finally, the effects of different temperature conditions are intriguing. What is the nature of the biophysical transformation of Sup35 in 30C cells compared to those exposed to 40C beyond that rather vague terminology of shift from SDS-resistant to SDS-sensitive*?

We have previously demonstrated that the prion domain Sup35 transitions from an SDS-sensitive form in its monomeric state to an SDS-resistant form in its amyloid state both *in vitro* (72) and *in vivo* (70). We have clarified this point in the text.

*What is special about 40C? Is this a useful tool to understand more completely the cellular environment*?

Based on our studies, 40°C-treated cells accumulate more heat-induced aggregates than cells treated at 37°C (Figure 3). This increase in accumulation correlates with an increase in the Hsp104 interactome (Figure 4), an increase in asymmetric localization of Hsp104 (Figures 4 and 5) and an increase in [*PSI*^*+*^]^Weak^ curing (Figure 1). Thus, the more severe thermal stress creates a unique proteostatic niche that allows Hsp104 to resolve amyloid *in vivo*. We have stressed these points in the revised manuscript both with the additional of new data and with modifications to the text.

*A broader context here would be useful as a number of laboratories have made observations on Q-bodies (*[22]*), iPOD, JUNQ (*[37]*), aggresomes (*[34]*), and spatial sequestration and symmetrical inheritance in yeast (*[76]*) have demonstrated that aggregates have spatial restrictions and numerous papers have shown that multiple chaperones are associated with these aggregate structures using a wide range of proteomic and co-localization methods, often with substantial consequences on cellular activity. Some of these would seem to have opposing outcomes, which then poses questions how the mother-daughter cell relationship of yeast relates to metazoan cell division, and broader relevance to neurons being post-mitotic*.

Please see the response to point 3 under “Response to points common to at least two reviews” above.

*Additional comments*:

*1) Evidence for involvement of Hsp104 in Sup35 amyloid disassembly comes both from the literature and new data presented here, that (i) the GdnHCl curing of [PSI+]weak and a ∼50% decrease in curing efficiency was observed in a heterozygous deletion of Hsp104 (*Figure 2*), (ii) Hsp104 was asymmetric localized upon thermal stress (*Figures 4 and 6*) and (iii) no significant changes were observed for Hsp104, Ssa1 and Sis1 chaperone expression levels upon heat stress (*Figure 3*). This establishes necessity, but not sufficiency and therefore does not rule out other 'factor(s)' induced upon heat stress that could contribute to amyloid disassembly*.

*These arguments can be strengthened by: (a) demonstrating whether the expression of other chaperone candidates (ie., other Hsp70 and Hsp40 isoforms) are affected by these thermal stress conditions. For example, there could be small changes in the expression of a number of chaperones that affects the cellular environment and enables amyloid resolubilization. Examples of candidates to test include chaperones examined in studies by*
[7]*;*
[75]*;*
[17]
*and*
[94]*, and (b) showing the over-expression data, currently not shown (p.18) that establishes that the o/e of Hsp104 alone, to the same level achieved in this study through its asymmetric retention, suffices to induce curing*.

We completely agree that our studies demonstrate necessity of Hsp104 asymmetric localization and activity for [*PSI*^*+*^]^Weak^ curing. We believe that establishing these points represents a significant advance in the field because up until this point, the engagement of chaperones with substrates has been correlated with reduced, not enhanced, activity. As we address in the Discussion, we believe that our system, which retains all factors within their native balance, is a major reason for this distinction.

We also agree that there are likely to be other factors that are necessary to promote Sup35 amyloid disassembly in response to thermal stress. In this manuscript, we focused the assessment of this possibility on factors that are known to be required for Hsp104-mediated fragmentation of Sup35 amyloid *in vivo* (Ssa1 and Sis1). We find no evidence for asymmetric localization of these factors (Figure 5—figure supplement 1).

With regard to curing by overexpression of Hsp104 alone, we have previously published this result for [*PSI*^*+*^]^Strong^ (15). For [*PSI*^*+*^]^Weak^, please see the response to point 2 under “Response to points common to at least two reviews” above.

*2) The authors' evidence for the formation of heat-induced, non-prion aggregates that accumulate upon heat stress and sequester Hsp104 in the mother cell rests on SDD-AGE (*Figure 1*), differential centrifugation (*Figure 3*) and that formation of aggregates correlates with other Hsp104 specific observations*.

*The protocol used for differential centrifugation is not sufficiently well described. It appears that the aggregates were quantified using a Bradford assay after pelleting in a tabletop centrifuge and a mild detergent wash. If the aggregates were not denatured before quantification, the presence of large particles may cause artifacts while making colorimetric measurements in a spectrophotometer. Since the quantification of these aggregates is an important aspect of this story, the authors should quantify aggregate formation with more vigor, using alternative procedures such as sucrose-density gradients and ultra-centrifugation to separate aggregates from other cellular debris. More central, the differential accumulation of aggregates in mother vs. daughter cells should be quantified and compared to the values on the differential retention of Hsp104*.

We have clarified our protocol for quantifying aggregates. While we did not denature the proteins before Bradford analysis, our protocol does include a preclear step to remove cellular debris. In lieu of additional characterization of aggregates on their own, we have instead included a more rigorous assessment of the Hsp104 interactome by co-immunocapture, SDS-PAGE and general protein staining. This experiment leads to the same conclusion: Hsp104 interacts with more proteins following a 40°C thermal stress than following a 37°C thermal stress (Figure 4).

We have not quantified the differential retention of aggregates following thermal stress because Hsp104GFP has been previously demonstrated to co-localize with model proteins (VHL, ubc9ts), which misfold upon thermal stress (37), and with oxidatively damaged proteins, which are asymmetrically retained in mother cells (21). However, we now show that the asymmetrically retained Hsp104GFP foci co-localize with a model substrate firefly luciferase-mCherry upon thermal stress (Figure 4—figure supplement 1).

*3) An important conclusion is that the heat-induced aggregates directly retain Hsp104, but there is no evidence for the direct interaction between these two factors, other than strong correlation. This would be greatly strengthened by an experiment that isolates these aggregates (see point 2 above) and demonstrates the presence of Hsp104 bound to these aggregates, using a digest and mass spectrometric quantification, for example. Alternatively, the authors could perform a co-IP experiment if they had more information on one of the proteins that is a constituent of these heat-induced aggregates*.

Please see the response to point 1 under “Response to points common to at least two reviews” above.

*4) Have the authors ruled out an effect on protein synthesis? Heat shock, in particular, is well known to inhibit protein synthesis, which could affect the flux, and shift equilibria in the proper cellular environment towards dissociation*?

There are two pieces of evidence, included in our original submission, that argues against this possibility. First, Sup35 protein levels are unchanged by the thermal stress (Figure 3—figure supplement 1), a point that we now clarify in the text. Second, treatment with cycloheximide alone does not lead to Sup35 solubilization (Figure 1), a point that we now highlight in the text.

*5) A point is regarding the relevance of the conclusions the authors make on the existence of cell-based limitations that preclude amyloid resolubilization in vivo to disease conditions. Since it is clear from their data that the cell-cycle stage is an important determinant that affects the success of resolubilization in vivo, this would not influence our understanding of the PQC in non-dividing neuronal cells. The authors should place their results in the broader context of protein misfolding diseases in the discussion*.

Please see the response to point 3 under “Response to points common to at least two reviews” above.

*6) Another point in the discussion on spatial engagement of PQC factors and effects on proteostasis capacity is a recent observation (*[99]*) on aggregate-associated sequestration of Hsc70 leading to down regulation of clathrin-mediated endocytosis. Does this suggest that the interactions of chaperones with aggregates could have different outcomes*?

The reviewer raises an interesting point. It is certainly possible that sequestration of chaperones can lead to different outcomes in different situations. Another possibility is that the outcomes are dictated by the experimental conditions. Our studies are distinct from those of Yu *et al.* in that the latter study employs aggregation-prone proteins that are expressed to high levels, which could promote imbalance in the system, as we address in the Discussion. We are in the early stages of understanding non-transcriptional mechanisms for regulating chaperone activity in cells, and these distinctions highlight interesting lines of investigation to pursue in the future.

Reviewer #3:

*1) 'but a direct demonstration of amyloid resolubilization in vivo has yet to be reported in any system'. Serio and colleagues have themselves already convincingly demonstrated amyloid resolublization by Hsp104 in vivo in their 2011 NSMB paper (*[15]*).*
[61]
*also provide compelling evidence that Hsp104 promotes amyloid solubilization in vivo. Moreover, there are numerous examples of amyloid clearance in conditional animal models of neurodegenerative disease models (e.g. Yamamoto et al. Cell 2000; Lim et al., J. Neuorosci. 2011). Hence, this statement is incorrect and misleading*.

We apologize for the confusion regarding this sentence. The text in question was meant to refer to studies in which chaperones were specifically overexpressed in amyloid model systems. We do agree with the reviewer that additional literature demonstrating amyloid clearance upon repression of synthesis of the amyloidogenic protein, as well as the effects of dominant-negative mutants are important to consider as well. We have re-written this part of the Introduction to highlight the facts that 1) chaperone overexpression has not been demonstrated to resolve amyloid *in vivo* and 2) amyloid clearance mechanisms must exist *in vivo*, although they are ineffective against continuously expressed, wildtype amyloid proteins.

With regard to the studies of Park *et al.* 2014, we disagree with the reviewer on the strength of the evidence supporting amyloid resolubilization *in vivo*. First, these studies are conducted under conditions of continued protein synthesis and over extended time frames, and as we have previously demonstrated (Satpute-Krishnan and Serio, 2005); newly-made protein quickly accumulates in a soluble form when Hsp104 activity is inhibited. Second, the interpretation of these microscopy experiments is inconsistent with previous biochemical studies that demonstrate an increase in the size of Sup35 aggregates upon Hsp104 overexpression, which we have reproduced for both the [*PSI*^*+*^]^Strong^ and [*PSI*^*+*^]^Weak^ variants (Figure 3—figure supplement 1). We have clarified these points in the Discussion.

*2) 'Rather, the effects of chaperone overexpression have been shown in some cases to be independent of their catalytic function (*[5]*,*
[6]*,*
[33]*,*
[61]*)'. This statement is a little misleading as in several cases the effect has been shown to depend on catalytic function (e.g.*
[12]
*PLoS Genet*.*)*

Again, we apologize for the confusion. Our statement was meant to only refer to the citations listed (hence the “some”), but we can see how this narrow discussion of the literature can lead to confusion. Rather than expand to include the Cushman-Nick study, which incidentally shows that overexpression of Hsp104 promotes amyloid formation by MJD rather than clearance of existing amyloid, we have deleted this sentence.

*3) 'Extracts from* C. elegans *and mammalian tissues and cell lines similarly promote amyloid solubilization (Cohen et al., 53 2006, Murray et al., 2010).' This statement should also be revised since Murray et al. subsequently published (Protein Sci. 2013 Nov;22(11):1531-41.) that 'our interpretation of the kinetic fibril disaggregation assay data previously reported in Bieschke et al., Protein Sci 2009;18:2231-2241 and Murray et al., Protein Sci 2010;19:836-846 is invalid when used as evidence for a disaggregase activity.*'

We thank the reviewer for pointing us to the Murray et al. 2013 manuscript. We have removed the previous references to disaggregase activity in metazoan extracts.